# HOW HARD ARE COMPUTER VISION DATASETS? CALIBRATING DATASET DIFFICULTY TO VIEWING TIME

## ABSTRACT

Humans outperform object recognizers despite the fact that models perform well on current datasets. Numerous attempts have been made to create more challenging datasets by scaling them up from the web, exploring distribution shift, or adding controls for biases. The difficulty of each image in each dataset is not independently evaluated, nor is the concept of dataset difficulty as a whole well-posed. We develop a new dataset difficulty metric based on how long humans must view an image in order to classify a target object. Images whose objects can be recognized in 17ms are considered to be easier than those which require seconds of viewing time. Using 133,588 judgments on two major datasets, ImageNet and ObjectNet, we determine the distribution of image difficulties in those datasets, which we find varies wildly, but significantly undersamples hard images. Rather than hoping that distribution shift or other approaches will lead to hard datasets, we should measure the difficulty of datasets and seek to explicitly fill out the class of difficult examples. Analyzing model performance guided by image difficulty reveals that models tend to have lower performance and a larger generalization gap on harder images. Encouragingly for the biological validity of current architectures, much of the variance in human difficulty can be accounted for given an object recognizer by computing a combination of prediction depth, c-score, and adversarial robustness. We release a dataset of such judgments as a complementary metric to raw performance and a network's ability to explain neural recordings. Such experiments with humans allow us to create a metric for progress in object recognition datasets, which we find are skewed toward easy examples, to test the biological validity of models in a novel way, and to develop tools for shaping datasets as they are being gathered to focus them on filling out the missing class of hard examples from today's datasets. Dataset and analysis code can be found at https://github.com/image-flash/image-flash-2022.

## 1 INTRODUCTION

Numerous efforts exist to build better evaluations for object recognizers. Broadly, these fall into four categories. Those that probe distribution shift, like ImageNetV2 (Recht et al., 2019). Those that add scale like OpenImages (Kuznetsova et al., 2020). Those that explicitly attempt to make images more difficult for models by adversarially selecting them, like ImageNet-A (Hendrycks et al., 2021) or adding artificial corruptions, like ImageNet-C (Hendrycks & Dietterich, 2019). And those that attempt to explicitly control for biases like ObjectNet (Barbu et al., 2019). These are responses to the fact that performance on standard benchmarks does not translate well to real-world conditions; 90% accuracy for one class in ImageNet does not mean that the detector will achieve 90% accuracy for that class in one's home or on frames of a movie. In all four cases, these efforts have no objective guide, no metric that evaluates how far they have progressed towards enabling models to generalize.

We set out to measure an orthogonal quantity – how difficult images in these datasets are for humans. Distribution shift and bias control won't on their own address this problem if datasets are overwhelmingly easy compared to what humans are capable of recognizing. And while scale helps, if datasets are heavily skewed toward images that are easy for humans, the statistics of performance on such datasets may hide the real underlying performance trends of models on harder images.

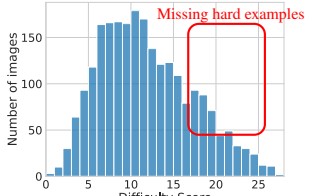 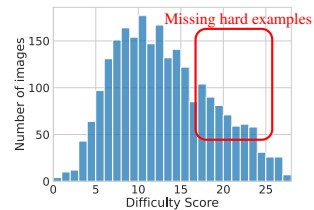

(a) ImageNet image difficulty distribution    (b) ObjectNet image difficulty distribution

Figure 1: The image difficulty distribution in ObjectNet and ImageNet. Difficulty here is defined as how many participants failed to recognize a given image across viewing times; easy images were almost always recognized even at short viewing times, while hard images were rarely recognized at short presentation times. Note that the difficulty of both datasets is roughly the same, and that hard images are under-sampled. Compared to what the human visual system can recognize, ImageNet and ObjectNet largely only test what can be recognized with short viewing times.

An objective metric by which to measure the difficulty of computer vision datasets has several advantages. First, we can determine if there are gaps in our datasets; perhaps certain difficulties are systematically undersampled. We find that this is the case: hard images are essentially missing. Moreover, merely aiming for distribution shift, even by changing how the dataset is gathered doesn't meaningfully change the difficulty distribution; ObjectNet and ImageNet were gathered from different sources (captured by Mechanical Turk vs the web), with different goals and additional controls for ObjectNet, yet their difficulty distributions are remarkably similar. Second, we can evaluate model scaling as a function of difficulty. We find that most model families scale poorly, performing well on the easy images but hardly improving on the hard images, with the exception of CLIP (Radford et al.). Third, it provides a new kind of metric for biological plausibility, orthogonal to raw performance – the error distribution, or how well networks predict neural activity. If a network is to be a model of the human visual system, not just an engineering model, then some quantity computed from that network should explain the observed difficulty scores. About half of the variance in the difficulty results is accounted for by a combination of c-score (Jiang et al., 2021), prediction depth (Baldock et al., 2021), and adversarial robustness (Goodfellow et al., 2015). Fourth, tools that measure difficulty could be incorporated into dataset collection and into how we report datasets and the overall progress of our community. In the long term, we intend to establish a dashboard giving a perspective on object recognition from the point of view of difficulty.

To build this difficulty metric, we choose as a proxy the minimum viewing time (followed by a backward mask) that a human viewer requires before being able to recognize the object in an image. Earlier readout is likely an indication that fewer mental resources were needed to recognize the image. After viewing the image, subjects have unlimited time to respond to a 1-out-of-50 forced choice task where they must identify the object class in the image that was shown. This metric is related to object solution time (OST) (Kar et al., 2018) explored in the neuroscience literature. We are of course not the first to carry out such viewing time experiments (Rajalingham et al., 2018). But, we do so at scale, with images from modern datasets, turn these results into a difficulty metric with practical applications, then predict this difficulty metric from quantities computed from current networks, and show the scaling of current models. We hope that in the future, benchmarks will regularly report their difficulty distribution (they can do so for only a few hundred dollars with the tools we provide) and that collections of benchmarks will seek out datasets based on their difficulty distribution. Additional attention may need to be paid while collecting datasets to not eliminate hard examples; any quick consensus-based process with multiple annotators is likely to be heavily biased against including hard examples. Practically, when collecting datasets for domains where the cost of errors is high (the medical domain, the military, etc.), being mindful of the difficulty distribution and actively shaping it to fill out harder images may be critical to building confidence in the resulting models.

Our contributions are:

1. a dataset of 133,588 human object recognition judgments as a function of viewing time for 4,771 images from ImageNet and ObjectNet,
2. the distribution of image difficulties for ImageNet and ObjectNet relative to what humans can recognize, shown in fig. 1,

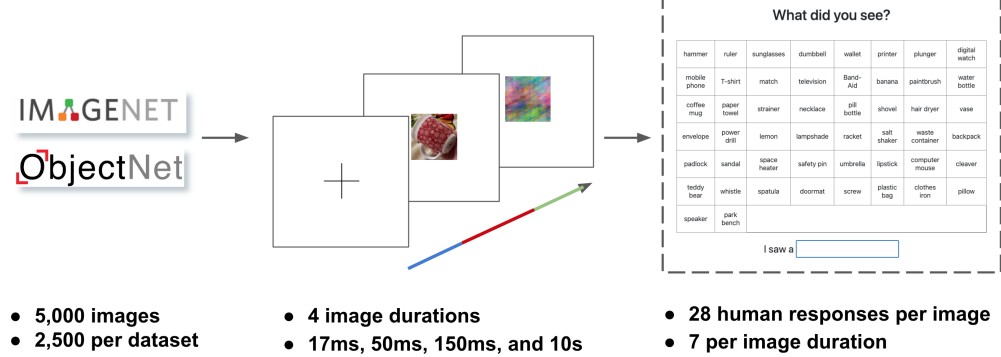

- **5,000 images**
- **2,500 per dataset**

- **4 image durations**
- **17ms, 50ms, 150ms, and 10s**

- **28 human responses per image**
- **7 per image duration**

Figure 2: Overview of the experiment. (left) 50 images from 50 object classes were randomly selected from both ImageNet and ObjectNet to total 5,000 images; of which we analyze 4771. Images were cropped in a square around the object of interest and then shown to human subjects on Amazon Mechanical Turk and in a controlled laboratory setting. (middle) Participants first saw a fixation cross for 500ms, then the image for either 17ms, 50ms, 150ms, or 10s, followed by a mask. After each image, subjects were given a 1-out-of-50 forced-choice task to identify the correct object class. (right) Each image was seen by 28 subjects, seven for each of the four image durations. No subjects saw the same image twice.

3. a new analysis of model performance as a function of image difficulty,
4. a new metric for validating the biological plausibility of object recognizers, predicting image difficulty, and
5. a new subset of images from ObjectNet and ImageNet sorted by difficulty for use in neuroscientific and behavioral experiments.

## 2 EXPERIMENT

We performed an experiment with human subjects on Mechanical Turk and in the lab in order to determine the minimum amount of viewing time required before subjects could identify an object present in an image. See fig. 2 for details. In what follows, we describe the stimuli, procedures, and validation of the online experiment with in-lab experiments.

### 2.1 STIMULI

We selected 2,500 images from the ImageNet validation set (which contains 50 images per class; we selected all 50 for 50 classes) and 2,500 images from the ObjectNet dataset (ObjectNet is only a test set). These images were evenly distributed among 50 object classes shared between ObjectNet and ImageNet. Since ObjectNet was gathered by participants taking pictures of objects in their homes, all of the object classes are household objects (see right panel on fig. 2 for a full list of the object classes) and all participants were likely familiar with all object classes used. The 50 classes were hand-picked to minimize similarity between classes that could be confusing for experiment subjects.

At short presentation times, subjects do not have time to fixate on multiple locations. Off-center objects would be recognized by peripheral vision. To eliminate this effect, we cropped all images around the target object using the ImageNet validation bounding boxes and new bounding box annotations for ObjectNet. In each case, we produced a square 224 by 224 image, padding with black if needed. Details about cropping can be found in the appendix. Note that this eliminates clutter and focuses only on the appearance of the object; we discuss this limitation in the conclusion.

### 2.2 PROCEDURES

Subjects were consented and provided a warning about photosensitivity to flashes. Only subjects with normal corrected vision were included in the experiment. Written instructions and a sample video of the experiment were then provided. Subjects then read through a list of the 50 object categories.

The Mechanical Turk experiments had additional calibration steps to determine the user's screen size (utilizing the fact that credit cards have a standardized size throughout the world) and distance from the screen (using a blind spot test) Li et al. (2020). Images were all shown at a size that would take up 8 degrees of visual angle in the participant's field of view. Further details about the setup steps and calibration are available in the appendix.

For an overview of the procedures, see the middle panel of fig. 2. Each trial had a randomly selected presentation time, either 17 ms, 50 ms, 150 ms, or 10 seconds; timings were selected to account for a participant's use of a 60Hz screen and to probe fine-grained distinctions between image difficulty most apparent at short timings. In each trial, a fixation cross was shown for 500 ms followed by the image, for one of the four timings. Immediately after the image, a phase gradient backward mask (Breitmeyer, 2007) was shown for 500 ms to disrupt further processing of the image. Details of the mask generation are described in the appendix. Participants were then presented with a grid of the names of 50 object categories and were asked to click on the object that they saw.

After each presentation, participants had to choose the identity of the object that was presented. As this was a forced-choice experiment, subjects were told to select their best guess if they are unsure. The order of the object classes in the grid was randomized at each trial. Subjects were given an unlimited amount of time to make a choice, although this time was recorded and appears to be anti-correlated with performance (quick decisions were likely to be more accurate than slow ones).

Experiments were counterbalanced. In total, each image was viewed by the same number of subjects at the same number of viewing times. Any one subject did not see the same image twice at any duration. Each participant viewed 50 images, one from each class, 25 from ImageNet and 25 ObjectNet, evenly spread across the four viewing times. The order of images, viewing times, and classes were randomized for each participant. Participants were allowed to complete several experiments; for those who did, we ensured that they saw disjoint images each time. In total, 1,495 workers took part in the experiment. Subjects on Mechanical Turk were compensated at over \$10 per hour. In total, we collected 140,000 trials, after which we discovered that a small number of images (229 images) were either incorrectly annotated, incorrectly cropped, or workers circumvented our automated means to ensure that no two workers saw the same image twice. This resulted in 133,588 trials for 4,771 images (28 presentations of each image, each of 7 subjects seeing each image at one of four timings).

### 2.3 IN-LAB VALIDATION

Great variation exists between monitors and browsers, with some showing the same images for nearly twice as long at short presentation times. Subjects as well can vary in their attention or alertness. To account for this, we validated the results with experiments in the lab. These followed the same procedures described above using the same interface with a gaming monitor (27 inch LG-27GN950-B monitor, with a 1ms GTG response time), on a machine with an NVIDIA 3060TI, using Chrome 102, running at 144Hz. We recorded the screen at 200fps and found that the presentation times accurately reflected the time periods during which this monitor showed each image.

We selected 200 images from the 5,000 used in the previous experiment, evenly distributed across the 50 classes and the two datasets (ImageNet and ObjectNet). Each participant saw all 200 images, at random presentation times in a random order; they never saw the same image twice, and could only participate in the experiment once. Subjects were instructed to stay at a fixed distance from the screen. An experimenter was present throughout to ensure that subjects were not distracted. Lighting was kept constant by closing the blinds. In total, 12 subjects (6 male, 6 female) participated in this experiment, and as can be seen in the next section, there was widespread agreement between the in-lab and Mechanical Turk experiments. It took subjects just short of an hour to complete the 200 images. Subjects were MIT graduate and undergraduate students compensated \$20 for their participation.

## 3 RESULTS

We considered an image as recognized at some viewing time when half of the participants could classify it. Chance on the 1-out-of-50 task is 2%; even a single correct response is an indication that something could be recognized. Typical images by minimum presentation time are shown in fig. 3.

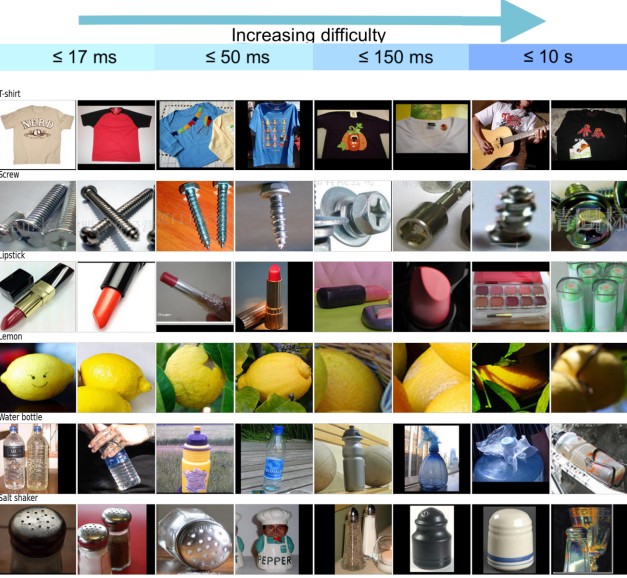

Figure 3: Images as a function of difficulty, using the minimum viewing time before they were reliably recognized, i.e., when more than half of subjects were correct. Harder images are more atypical, have more difficult lightning, more occlusions, and are sometimes more blurry. Additional examples are available in the appendix and online. Most datasets contain images like those recognized at short timings.

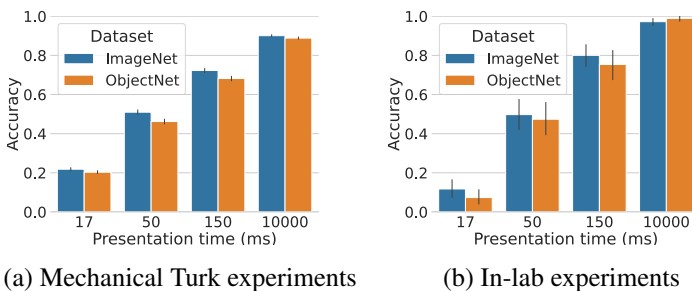

(a) Mechanical Turk experiments      (b) In-lab experiments

Figure 4: Accuracy as a function of presentation time. The same images (5,000 online, 200 in lab) were presented at 4 timings. Results for both conditions were similar, although in-lab experiments achieved nearly 100% accuracy with 10 second viewing times, while halving the performance at 17ms compared to MTurk. The accuracy improvement at long times is likely because of the more controlled conditions with fewer distractors in lab. The lower accuracies at short timings are likely due to issues with displaying images at short timings: some monitors are very slow and can fade the image in and out effectively displaying it for twice the intended 17ms.

Images that are quickly recognized by humans, easy images, are most similar to those seen in current datasets, while more difficult images include occlusion or difficult lighting.

An overview of accuracy as a function of viewing time online and in the lab is shown in fig. 4. Both experiments broadly agree with one another. In-lab experiments have higher variances due to having 20x fewer images and half as many subjects per image. In lab, the performance on short timings was significantly worse, half of that seen online. We believe this is largely due to slow monitors which display the image for significantly longer than 17ms, roughly twice as long. Recording screens with high-speed cameras supports this hypothesis. At the high end, subjects in lab were nearly 100% accurate, 10% higher than online. We believe this likely has to do with how distracted subjects were. In what follows, we focus on the Mechanical Turk experiments due to their scale.

Human performance drops off steeply when difficult images are shown for shorter viewing times; see fig. 5. This makes the experiment sensitive to even minor variations in difficulty.

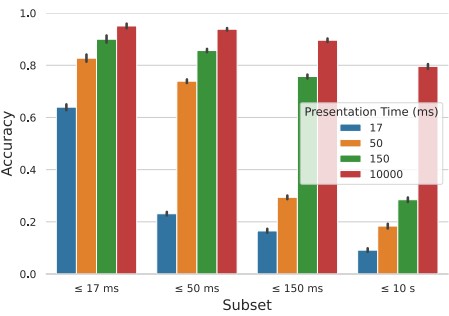

Figure 5: Human accuracy on Mechanical Turk as a function of the difficulty subset. Colors denote presentation time, while the x-axis denotes subsets of images that require at least the stated viewing time to be recognized reliably. Accuracy drops off steeply when hard images are displayed at shorter time intervals.

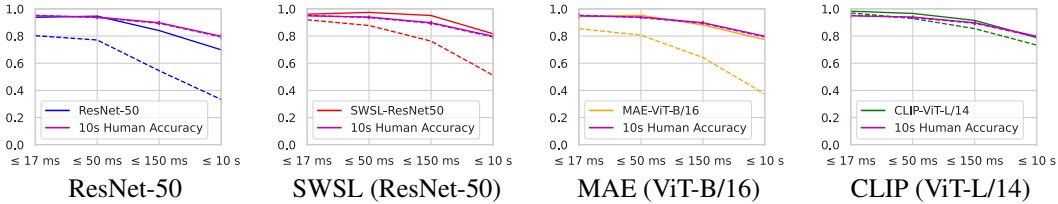

Figure 6: Accuracy of four recent models on ImageNet (solid) and ObjectNet (dashed) as a function of image difficulty (extensive experiments with numerous additional models are available in the appendix). Note that this understates human performance, as it includes the Mechanical Turk, not the in-lab experiment results. Even for the hardest images, humans have nearly perfect performance in lab. Model performance drops off significantly for the harder images. Harder images also show a much larger gap when testing on ObjectNet.

### 3.1 HOW HARD ARE TODAY'S OBJECT RECOGNITION DATASETS?

The minimum viewing time required for reliable recognition is a proxy for image difficulty. Datasets today are not gathered to control for difficulty and, indeed, when plotting the difficulty of images in ImageNet and ObjectNet, we find that the difficulty curve for these datasets is highly skewed; see fig. 1. Rather than plotting four bins, one for each viewing time, we plot a more fine-grained quantity: the total number of incorrect responses out of the 28 presentations of each image (7 participants at 4 timings). Images with few incorrect responses are easy: all participants at all timings could recognize them, even the short timings. Images with many incorrect responses are hard: few participants could recognize the images at only a few longer timings.

Hard images are vastly underrepresented in today's datasets. The difficulty distribution of ImageNet and ObjectNet are very similar to one another. This is despite the fact that the latter was collected in a manner intentionally intended to highlight more diverse and less biased images. Merely collecting more images does not appear to help. Instead, we must develop new tools to guide dataset collection toward more difficult exemplars; we describe such a tool and workflow below.

### 3.2 WHAT WE CAN LEARN ABOUT MODELS FROM HARD IMAGES

Machine accuracy varies as a function of the minimum viewing time required to recognize the object in an image. Most models, but not all, see a significant performance dropoff between the easy images and the hard images, see fig. 6. See the appendix for extensive results for dozens of models broken down by image difficulty. Note that this understates human performance as it shows the Mechanical Turk results. In-lab, even for the hardest images, humans have nearly perfect performance.

These results also show that the gap between ImageNet and ObjectNet performance increases as image difficulty increases. Likely, many phenomena such as distribution shift are much more acute for harder rather than easier images. Datasets intended to challenge object recognizers would benefit from sampling hard images, rather than vastly oversampling easy images as they do today.

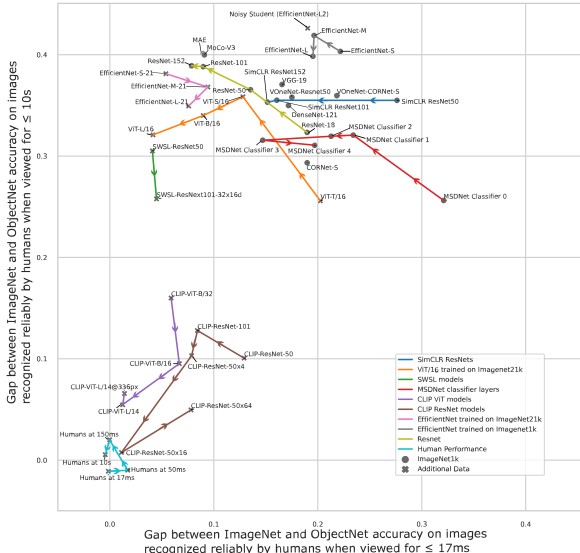

Figure 7: Model robustness trends as a function of image difficulty. The horizontal axis shows the accuracy gap between ImageNet and ObjectNet for easy images. The vertical axis shows the accuracy gap between ImageNet and ObjectNet for hard images. Each point is a model. Arrows connect models that are part of the same family, starting with smaller models pointing to larger ones. Points on the diagonal have the same gap in performance between ObjectNet and ImageNet. Humans hover around zero with some small noise, they are robust with respect to the distribution shift between ObjectNet and ImageNet. Models that are part of the same family with increasing number of parameters are connected by lines. As model size increases, plain ResNets improves on easy images, but increase their gap on harder images. SimCLR with increasing model size is horizontal, it narrows the gap on easy images, but hardly improves on hard images. CLIP stands out. Larger CLIP models tend toward having little to no gap between ImageNet and ObjectNet, although, this is not monotonic and appears to reverse itself as the network grows in size.

Image difficulty can tease apart differences between recognition models that would otherwise be lost because of the skewed underlying difficulty distributions in current datasets. In fig. 7, we plot numerous object recognition models contrasting their performance on the easiest and hardest images. Rather than computing the absolute performance of models, that would naturally favor larger models with larger training sets, we measure the gap in performance between ImageNet and ObjectNet. A more robust detector is one that has a smaller gap, even if its performance is lower, as scaling models up (both in parameter size and training set size) is well understood. Models that are part of the same family are connected by arrows starting with the smaller variants pointing to the larger ones.

Humans hover around zero; they are robust with respect to the distribution shift between ObjectNet and ImageNet. Some model families are horizontal, like SimCLR, or even have a negative slope. This shows that as they scale, their performance is increasing on the easy images, but not on the harder images, or in the case of negative slopes, it is widening the gap on harder images. CLIP stands out, and most closely approaches the human results. Extrapolating the performance of models on such a graph shows that many current model families are likely to stop or radically slow down performance improvements on hard images, a much less optimistic story than if one merely considers aggregate performance. At the same time, CLIP shows promise, although it too appears to eventually begin to move away from being robust to distribution shifts as model size increases. Note that these experiments use the largest CLIP models available to the public; other large models described in the CLIP publication may have even more optimistic results.

## 3.3 EXPLAINING IMAGE DIFFICULTY AND TESTING HOW SIMILAR MODELS ARE TO HUMANS

If models are to not just perform well, but to also process images in ways that are similar to how the human visual system does, then they should contain a proxy for difficulty. Most models today are not recurrent, so no direct analog exists to viewing time (although, even if they were recurrent, it is unclear if directly taking the number of iterations of the model is the correct analog of viewing time).

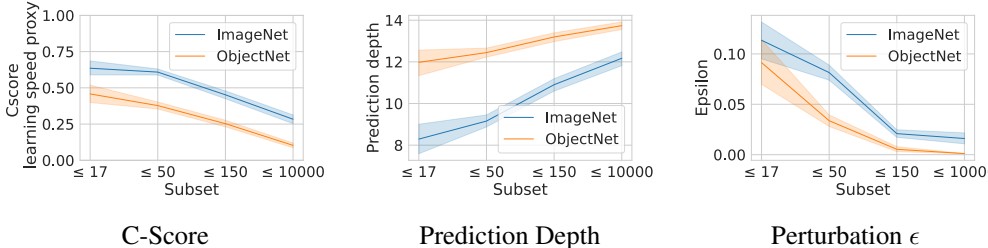

Figure 8: The correlation between three metrics and image difficulty. On the x axis images subset by the minimum amount of viewing time, in ms, required for the majority of participants before they are recognized. All three metrics are correlated with difficulty. Hard images are learned later in model training, predicted by later layers, and need much smaller perturbations to attack.

Minimum viewing time judgments provide an independent and complementary metric by which to evaluate how similar models are to human brains, in addition to their behavior, error pattern, and ability to explain neural recordings.

We investigate three quantities computed from models that could be used to explain the difficulty judgments: c-score (Jiang et al., 2021), prediction depth (Baldock et al., 2021), and adversarial robustness (Goodfellow et al., 2015). C-score is a consistency score computed while the network is training; we use the learning speed proxy to compute it. Prediction depth measures the earliest layer at which a network produces its final prediction. And adversarial robustness refers to the perturbation, $\epsilon$, needed to fool the network on a given example. See fig. 8 for an overview of the correlation between these metrics and difficulty. The same network with a ResNet-50 architecture trained on ImageNet was used for all experiments presented here. Note that $\epsilon$ is only computed over the correctly classified images.

This analysis reveals that images that require more viewing time for humans are harder for networks in several ways. They are learned much later in the training process. They are predicted by later layers in the network. And they require much smaller perturbation to create adversarial examples. This underscores that human minimum viewing time has many practical consequences for how networks process images. At the same time, these metrics alone do not explain all of the difficult judgments. Logistic regression including all three metrics classifies images into the four difficulty subsets with 48.5% accuracy, far above chance, but leaving plenty of room to do more. New metrics for understanding how networks process images could be validated against this data.

## 4 RELATED WORK

**Image difficulty** Image difficulty has previously been investigated primarily from the perspective of models. Jiang and Zhang et al. constructed a metric c-score (Jiang et al., 2021), to measure how much a model was required to memorize to learn a particular image instance. It also proposed a learning speed proxy for image difficulty; more difficult images are classified correctly later in the course of training. Baldock et al. measured image difficulty by measuring the model layer depth at which a model makes its final prediction for images (Baldock et al., 2021). More difficult examples can only be decoded to the model final prediction from later layers. Agarwal et al. introduced the variance of gradients (VoG) difficulty metric (Agarwal et al., 2022). VoG is computed over the course of training, similar to the c-score learning speed proxy, but it uses the variance in the gradient updates over training instead of the models final prediction. More difficult examples have a higher variance. Since each of these methods depends on model performance rather than human performance they find ObjectNet significantly more difficult than ImageNet, unlike our human presentation time metric as show in fig. 1.

**Dataset design** As a field, we have measured our progress by models' performance on tests created by splitting a random subset of images from large-scale image datsets. Almost all of the most influential datasets were created by web scraping publicly-available images and labeling them based on a consensus among human annotators (Russakovsky et al., 2015). More recently, researchers have realized that there is significant value to testing models' abilities to recognize objects out of

domain. This has included attempting to recreate existing test sets from new data, ImageNetV2 (Recht et al., 2019), constructing adversarial test sets for models, ImageNet-A (Hendrycks et al., 2021), as well as adding corruptions to existing test sets, ImageNet-C (Hendrycks & Dietterich, 2019). The ObjectNet (Barbu et al., 2019) test set took this one step further by collecting a new set of images sampling uniformly across a set of visual controls. In this work we find that the viewing time difficulty distributions between ImageNet and ObjectNet are similar, but models demonstrate a larger robustness performance gap on hard images.

**OOD model performance** Prior work studying object recognition models performance has found a linear trend in performance improvement on ImageNet and other OOD datasets (Recht et al., 2019; Taori et al., 2020; Barbu et al., 2019). This linear trend can be beaten slightly when massively scaling up the quantity of training data used. Recently, the multimodal models CLIP (Radford et al.) and LiT (Zhai et al., 2022) have demonstrated a break from this linear trend, greatly increasing out of distribution generalization performance by using multimodal learning.

**Limited viewing time** Geirhos 2018 et al. (Geirhos et al., 2017) studied human and machine performance under limited presentation time with test set images made more difficult by adding various image corruptions. Finding that humans are far more robust to these difficulty changes than machines. Rajalingham and Isaa et al. (Rajalingham et al., 2018) studied human and primate object recognition performance under limited presentation time by presenting 100ms images finding models unable to match image-level behavioral patterns of primates.

## 5 Conclusion, limitations, and future work

Rather than focusing on scaling, distribution shift, or control for biases alone, we should also focus on dataset difficulty explicitly. Today's datasets skew toward being too easy by undersampling hard images. ObjectNet was designed for distribution shift and bias control and was not collected from the web, yet its distribution of image difficulties is remarkably similar to that of ImageNet. By focusing on ways to measure dataset difficulty as datasets are collected, we can better calibrate the entire community, and create the resources needed to push object recognition forward. In addition to just creating better datasets, understanding performance as a function of difficulty reveals radically different scaling curves for different models and approaches. It can also provide subsets of images that highlight different types of processing for neuroscientific or behavioral experiments.

Our approach to measuring the difficulty of object recognition tasks focuses specifically on object instances, rather than other factors that also add complexity such as saliency and clutter. By cropping images around the object, we make figure-ground segmentation far easier and separate out the issue of the difficulty of this specific instance of an object as opposed to the difficulty of finding this object within a cluttered background.

The particular notion of image difficulty derived from this experiment is just one metric of difficulty. As we describe in the introduction, other metrics exist which can be computed automatically given models that recognize objects. But those metrics rely on the models and change as models change. The metric we present here is both absolute, it is calibrated to what humans can achieve, and model-agnostic. Models change over time, far faster than datasets do, there is a real danger when designing datasets that their difficulty will be tuned to specific models, which can be avoided by using measures related to humans.

While we performed extensive experiments to validate the approach presented here, that having been established, measuring the difficulty of any one dataset is now easy and cheap. One can sample a few hundred images and run an experiment on Mechanical Turk. This only costs on the order of hundreds of dollars per dataset and can be carried out quickly. The more critical the dataset, and the cost of object recognition failures, the more important it is to do so. Of course, while doing so experimenters should ensure that participants are treated fairly and paid appropriately.

Roughly 4 weeks of compute time on 2 machines with 8 TITAN RTX were used to generate the results, largely in computing c-score, prediction depth, adversarial robustness, and finetuning models. We release our experiments under the Creative Commons Attribution 4.0 license.

This paradigm for measuring difficulty could be adapted to other vision tasks such as segmentation or optical flow. Other tasks are out of scope, like visual search, as they literally require multiple saccades. Calibrating our field to what humans can do across a wide range of tasks, datasets, conditions, remains a significant challenge, but one that we think can now be addressed.

ETHICS STATEMENT

We received an IRB exemption for this study as it poses a minimal risk to participants. No personally identifiable data was collected from any subject. Both in person study participants and Mechanical Turk participants were compensated appropriately.

We hope that more objective measures of dataset difficulty will lead to more dataset equity. As it appears that gathering more difficult datasets requires more deliberate effort, an objective metric could hold creators to account. For example, datasets for training autonomous cars could represent different socioeconomic conditions equally, but more time might be invested in gathering higher-quality data in more affluent conditions. This can now be brought to light by computing the difficulty of each subset of the data, ultimately we hope leading to fairer datasets.

REPRODUCIBILITY STATEMENT

We publicly release all data and analysis code used in this study at the URL https://github. com/image-flash/image-flash-2022. This includes the responses from all experiment trials used in our analysis and code for reproducing the results and figures presented in this paper. We also included scripts for ImageNet and preprocessed images for ObjectNet to reproduce the stimuli images used in this study.

We provided a detailed description of our studies procedures, including screenshots, in the main text and the appendix. After publication, we plan to fully release the web tool we developed for performing these experiments.

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

# A  APPENDIX

## A.1  IMAGE CROPPING PROCEDURE

1. We draw a bounding box around the object (we use existing bounding boxes for the ImageNet validation set and collect our own bounding boxes for ObjectNet from MTurk).

2. We initialize the cropping box to be the bounding box.

3. If the cropping box does not form a square, we extend the shorter side of the rectangular cropping box to form a square. If the image is not large enough to extend the shorter side of the cropping box, we pad it with black pixels to form a square.

4. We crop using the cropping box for the image. The cropped image will be a square.

5. We resize the cropped image to be 224x224 pixels.

## A.2  MASK GENERATION

The masks were generated following the procedure used by (Kar et al., 2019). Specifically, a Fourier transform was applied to each image to obtain the magnitude and phase components. Then, a random array with elements sampled uniformly from [0, 1] was added to the image phase component after which the magnitude and phase components were recombined via an inverse Fourier transform to produce the mask. Each image was paired with its particular phase-scrambled mask in the experiments.

## A.3  EXPERIMENT PROCEDURE AND PAYMENT

Participants both in the lab and on Mechanical Turk were presented with a document informing them of the purpose, privacy, and risks associated with the experiment and soliciting their consent to participate (see fig. 9). Participants were then instructed as to how to carry out the experiment and were shown an example video as well as the list of image classes for their review before beginning. They were informed that they would not need to memorize the classes as the classes would be shown to them after each video. Participants were also encouraged to take breaks should they feel fatigued or otherwise uncomfortable. Example instructions are shown in fig. 10

After giving consent and reading the experiment overview. participants then completed two calibration steps for to estimate the size of their monitor and their distance from the screen for us to then size the videos appropriately to 8 degrees of visual angle. First, the participants are shown an image of a credit card and are asked to use a card of their own to adjust a slider to change the size of the card on the screen to the size of their card. Since credit cards are the same size around the world, this allows us to measure the pixel-to-inches ratio of the participant's monitor. Next, the participant completes a blind-spot test (Li et al., 2020) that allows us to estimate the distance they are sitting from their screen. Together, these two measurements are sufficient to compute the desired video eccentricity. See fig. 11 for images of the calibration steps.

The estimated hourly wage for participants on Mechanical Turk and in the lab was $10/hr and $20/hr respectively with approximately $15,000 spent in total on participant compensation.

## A.4  IN-LAB EXPERIMENT RESULTS

To corroborate our Amazon Mechanical Turk results, we selected 200 images shown to Turk workers to conduct the same experiment in a controlled laboratory setting. 12 individuals came to participate in the experiment in which they viewed and responded to all 200 images on our 144Hz refresh rate monitor with 1ms gray-to-gray time. After conducting the experiment, 3 individuals had seen each image at each of the 4 presentation times. When compared to the MTurk results for those same 200 images, the comparison is much as we would expect. The In-Lab accuracy with shortest image duration (17ms) is less than on MTurk which can likely be contributed to the use of our new, high refresh-rate monitor in the controlled environment. It is likely that MTurk workers' personal computers differ in their graphics presentation abilities which may result in the image being visible for slightly greater than 17ms on some monitors. On the other end, the in-lab experiments reported higher accuracy at the longest image duration (10s) which is also unsurprising as the in-lab participants

Table 1: Dataset statistics

| | |
|---|---:|
| Number of responses | 133,588 |
| Number of images | 4,771 |
| Number of presentaiton durations | 4 |
| Number of response per image | 28 |
| Number of ObjectNet images | 2,415 |
| Number of ImageNet images | 2,356 |
| Number of participants | 1,495 |

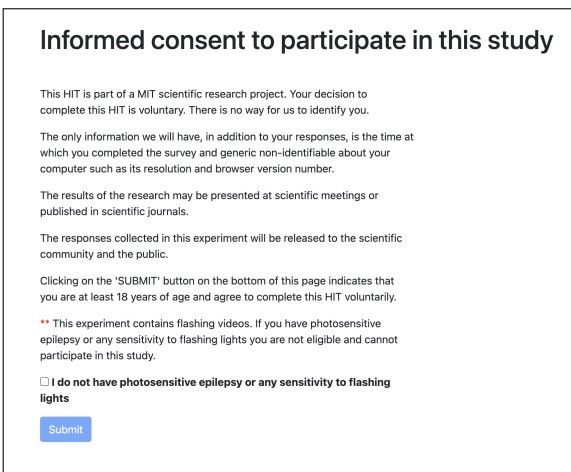

Figure 9: Informed consent page shown to participants before beginning the experiment.

completed the task in a controlled environment with no distractions and are likely more inclined to take the task seriously and stay focused. The results show no significant differences in accuracy at the intermediate image durations. See fig. 4 for a side-by-side comparison between MTurk and In-Lab results.

## A.5 DATASET STATISTICS

We collected 28 human responses for each of 5,000 images (2,500 from ImageNet and 2,500 from ObjectNet). After reviewing response, 229 images were removed due to either being unrecognizable, mislabeled, or having been seen by the same worker twice despite safeguards in place to disallow it. Additional dataset statistics are listed in table 1.

## A.6 FINETUNED MODELS

Here we list details regarding training/finetuning procedures for the model results reported in the paper.

### A.6.1 MODEL TRAINING PROCEDURE

Pretrained models weights were instantiated using publicly available model checkpoints, either using torchvision or found on the model's source repository. The models—with the exception of CLIP—were then finetuned using subsets of the ImageNet training and validation sets containing only the 50 classes we chose to use in the psychophysics experiments. The models were finetuned for 90 epochs with an SGD optimizer and initial learning rate of 0.1 with momentum value of 0.9 and weight decay coefficient of 0.0001. The learning rate decayed by a factor of 2 every 9 epochs.

## Task Overview

Please read the following:

You are participating in an experiment in which we are studying people's ability to recognize what object is in an image given varying amounts of image viewing time.

Included below is an example of what we will ask you to view. Look at the center of the cross. When the video plays, the cross will briefly change to an image of an object and then it will change to an image of a random pattern. After you see the image we will ask you what object you saw in the image. You will only be able to play the video once, so be ready and focused before pressing play. The image may disappear too quickly for you to immediately be able to name the object that you saw, that's okay. Look at the multiple choice list of possible objects, think about what you saw, and take your best guess. There is no time limit for your response, but you have to respond before you can move on to the next image.

**Play example video**

---

The multiple choice list of objects will be the same for all the images. Take a minute to read all these object names. If you are unfamiliar with a category or find it confusing, do a quick search on the web to see some example images. YOU DO NOT NEED TO MEMORIZE THESE CATEGORIES, they will be shown to you again after you watch each video.

| | | |
|---|---|---|
| backpack | banana | band-aid |
| bench | butcher's knife / cleaver | cell phone |
| computer mouse | doormat | power drill |
| envelope | hair dryer | hammer |
| clothes iron | lampshade | lemon |
| lipstick | match (i.e. matchstick) | mug |
| necklace | padlock | paintbrush |
| paper towel | pill bottle | pillow |
| plastic bag | plunger | portable heater |
| printer | ruler (i.e. measuring stick) | safety pin |
| salt shaker | sandal | screw |
| shovel | spatula | speaker |
| strainer | stuffed animal | sunglasses |
| t-shirt | racket (i.e. tennis racket) | waste container (i.e. trash bin) |
| television (TV) | umbrella | vase |
| wallet | watch | water bottle |
| dumbbell | whistle | |

---

On the response page, you will see a table with these 50 options displayed. The order of the options in the table will be randomized but all 50 options will always be visible. Please take the time to find the correct answer if you know what it is. If you do not know what you saw, take your best guess. **We will know if you are not trying to answer correctly and your submission will be rejected. Please take this task seriously.**

In this experiment you will look at some images and tell us what objects you saw. We expect this to take about 60 minutes. You can take breaks at any time. If you feel like you are tired or having trouble focusing please take a break and resume the experiment later. Please try to take your breaks after submitting a response for an image and before clicking play on the next image. Responses are saved as you submit them so as long as you return to the same link, you will be able to pick up where you left off.

We would also like you to view our experiment in a quiet and well-lit room if possible.

Thank you for participating

Figure 10: Instructions given to participants before beginning the experiment.

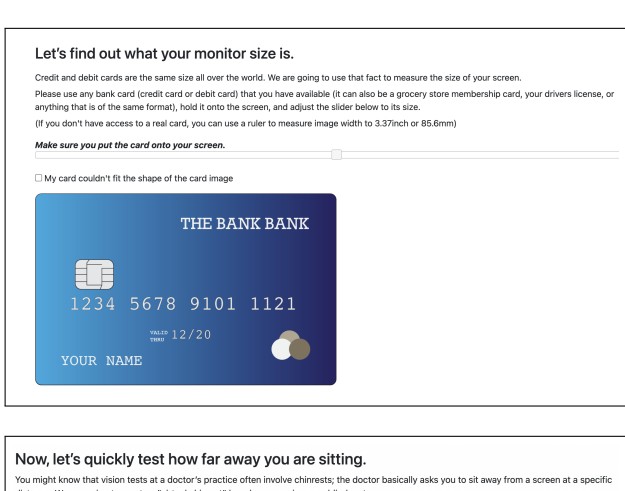

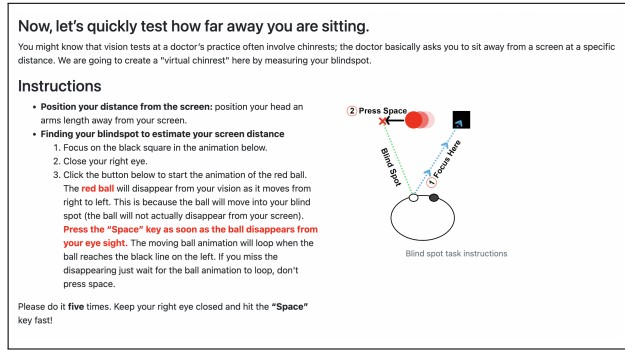

Figure 11: Images of the experiment calibration steps. The credit card task was used to measure the pixel-to-inches ratio of the subject's screen. The blind spot task provided an estimate of the subjects distance from their screen.

Table 2: Model accuracy per recognition time subset. Models are named to include architecture, training objective, and training dataset where appropriate. ResNet-X-Y% indicates a ResNet with depth X and trained on a random Y% subset of the ImageNet-1k dataset. Model names ending in 21k were pretrained on ImageNet-21k. All other models with the exception of SWSL and CLIP models were pre-trained on the full ImageNet-1k dataset.

| Subset | <= 17 | | <= 50 | | <= 150 | | <= 10000 | |
| Dataset | Image-Net | Object-Net | Image-Net | Object-Net | Image-Net | Object-Net | Image-Net | Object-Net |
|---|---|---|---|---|---|---|---|---|
| ResNet-18 He et al. (2016) | 92.7 | 73.7 | 89.8 | 63.0 | 76.2 | 41.7 | 55.7 | 23.3 |
| ResNet-18-80% | 87.6 | 66.4 | 85.0 | 60.9 | 71.2 | 38.3 | 51.6 | 20.8 |
| ResNet-18-60% | 84.8 | 67.2 | 85.7 | 57.4 | 70.2 | 38.1 | 51.1 | 20.8 |
| ResNet-18-40% | 83.1 | 58.4 | 83.1 | 52.0 | 63.5 | 36.5 | 48.7 | 18.8 |
| ResNet-18-20% | 78.7 | 46.7 | 74.1 | 44.5 | 60.7 | 26.0 | 38.3 | 13.3 |
| ResNet-50 He et al. (2016) | 93.8 | 80.3 | 94.6 | 77.1 | 84.0 | 54.5 | 69.9 | 33.3 |
| ResNet-50-80% | 88.2 | 75.9 | 89.9 | 68.7 | 80.6 | 48.3 | 62.9 | 26.3 |
| ResNet-50-60% | 89.3 | 70.1 | 89.0 | 67.4 | 75.6 | 46.7 | 61.4 | 25.1 |
| ResNet-50-40% | 92.7 | 68.6 | 86.2 | 59.5 | 72.3 | 40.5 | 53.7 | 23.3 |
| ResNet-50-20% | 85.4 | 58.4 | 81.1 | 48.4 | 66.8 | 32.9 | 46.3 | 19.2 |
| ResNet-101 He et al. (2016) | 94.4 | 85.4 | 94.3 | 78.8 | 88.6 | 59.6 | 74.7 | 35.9 |
| ResNet-101-80% | 92.1 | 74.5 | 92.5 | 72.9 | 81.9 | 51.7 | 65.5 | 30.0 |
| ResNet-101-60% | 91.0 | 75.9 | 88.3 | 67.1 | 78.8 | 46.7 | 61.7 | 27.6 |
| ResNet-101-40% | 89.3 | 67.9 | 87.9 | 60.0 | 74.8 | 40.2 | 56.9 | 22.2 |
| ResNet-101-20% | 85.4 | 51.8 | 83.1 | 50.5 | 66.6 | 34.4 | 51.3 | 17.1 |
| ResNet-152 He et al. (2016) | 93.3 | 85.4 | 95.5 | 80.9 | 89.9 | 62.9 | 75.2 | 36.3 |
| ResNet-152-80% | 90.4 | 75.2 | 90.6 | 71.3 | 82.7 | 51.1 | 67.2 | 29.2 |
| ResNet-152-60% | 92.7 | 73.0 | 89.0 | 66.8 | 79.2 | 48.8 | 61.9 | 27.3 |
| ResNet-152-40% | 92.1 | 65.7 | 86.8 | 61.4 | 73.9 | 43.5 | 56.4 | 22.9 |
| ResNet-152-20% | 85.4 | 60.6 | 81.3 | 50.8 | 69.5 | 35.1 | 48.9 | 16.3 |
| CORNet-S Kubilius et al. (2019) | 92.7 | 73.7 | 90.6 | 68.7 | 78.3 | 47.0 | 55.4 | 26.1 |
| VOneNet-Resnet50 Dapello et al. (2020) | 92.7 | 75.2 | 92.7 | 72.5 | 81.9 | 49.5 | 61.7 | 25.9 |
| VOneNet-CORNet-S | 90.4 | 68.6 | 90.0 | 63.2 | 78.8 | 44.1 | 58.3 | 22.4 |
| VGG-19 Simonyan & Zisserman (2015) | 91.0 | 74.5 | 88.6 | 65.7 | 78.2 | 45.9 | 60.0 | 22.9 |
| Noisy Student (EfficientNet-L2) Xie et al. (2020) | 94.9 | 75.9 | 92.5 | 66.6 | 85.3 | 45.8 | 67.7 | 25.1 |
| DenseNet-121 Huang et al. (2017b) | 93.8 | 76.6 | 92.0 | 73.0 | 82.2 | 50.9 | 63.6 | 28.6 |
| MSDNet Classifier 0 Huang et al. (2017a) | 78.1 | 46.0 | 73.3 | 38.2 | 55.7 | 27.2 | 38.6 | 12.9 |
| MSDNet Classifier 1 | 87.6 | 64.2 | 84.6 | 54.6 | 70.0 | 37.1 | 51.1 | 19.0 |
| MSDNet Classifier 2 | 89.9 | 68.6 | 88.7 | 62.2 | 73.8 | 45.3 | 57.8 | 25.9 |
| MSDNet Classifier 3 | 89.9 | 75.2 | 88.9 | 66.6 | 73.3 | 45.8 | 57.8 | 26.3 |
| MSDNet Classifier 4 | 92.7 | 73.0 | 89.5 | 67.8 | 75.2 | 45.6 | 58.3 | 27.3 |
| SimCLR ResNet50 Chen et al. (2020) | 88.2 | 60.6 | 84.5 | 59.3 | 70.0 | 46.1 | 56.9 | 21.4 |
| SimCLR ResNet101 | 92.7 | 76.6 | 88.3 | 68.8 | 80.0 | 52.9 | 64.3 | 28.8 |
| SimCLR ResNet152 | 93.3 | 78.1 | 90.2 | 70.0 | 81.4 | 52.7 | 67.5 | 32.2 |
| CLIP-ViT-B/32 Radford et al. | 94.9 | 89.1 | 88.8 | 78.2 | 75.7 | 60.7 | 56.4 | 40.4 |
| CLIP-ViT-B/16 | 97.2 | 90.5 | 93.2 | 86.2 | 81.3 | 73.1 | 62.7 | 53.1 |
| CLIP-ViT-L/14 | 98.3 | 97.1 | 96.6 | 92.9 | 91.5 | 85.4 | 78.8 | 73.3 |
| CLIP-ViT-L/14@336px | 97.8 | 96.4 | 96.3 | 93.6 | 91.2 | 87.5 | 79.5 | 72.9 |
| CLIP-ResNet-50 | 91.0 | 78.1 | 81.6 | 70.1 | 65.0 | 55.0 | 42.4 | 32.4 |
| CLIP-ResNet-101 | 93.8 | 85.4 | 85.4 | 75.1 | 70.0 | 58.4 | 49.6 | 36.9 |
| CLIP-ResNet-50x4 | 93.3 | 85.4 | 86.4 | 77.5 | 73.6 | 64.7 | 51.1 | 40.8 |
| CLIP-ResNet-50x16 | 93.8 | 92.7 | 89.8 | 83.2 | 78.3 | 72.5 | 53.5 | 52.7 |
| CLIP-ResNet-50x64 | 98.3 | 90.5 | 94.5 | 89.6 | 86.0 | 82.6 | 65.5 | 60.6 |
| EfficientNet-S Tan & Le (2019) | 89.3 | 67.2 | 92.0 | 63.8 | 78.0 | 44.3 | 63.9 | 23.5 |
| EfficientNet-M | 90.4 | 70.8 | 89.9 | 63.4 | 75.4 | 39.9 | 63.9 | 22.0 |
| EfficientNet-L | 93.3 | 73.7 | 92.8 | 68.2 | 83.9 | 47.4 | 68.7 | 28.8 |
| EfficientNet-S-21 | 96.6 | 91.2 | 95.2 | 80.7 | 89.1 | 63.7 | 75.2 | 37.1 |
| EfficientNet-M-21 | 97.8 | 88.3 | 96.4 | 84.7 | 90.1 | 64.6 | 77.6 | 40.8 |
| EfficientNet-L-21 | 96.6 | 89.1 | 96.6 | 84.1 | 91.2 | 67.0 | 78.1 | 43.1 |
| ViT-T/16 Dosovitskiy et al. (2021) | 64.0 | 43.8 | 69.1 | 40.5 | 54.2 | 25.7 | 37.3 | 11.8 |
| ViT-S/16 | 93.8 | 81.0 | 92.8 | 73.6 | 82.6 | 51.7 | 65.1 | 29.2 |
| ViT-B/16 | 94.4 | 85.4 | 95.3 | 76.3 | 85.2 | 59.3 | 66.7 | 32.7 |
| ViT-L/16 | 98.3 | 94.2 | 97.5 | 90.4 | 96.4 | 79.4 | 83.9 | 51.8 |
| MAE He et al. (2022) | 94.4 | 85.4 | 95.3 | 80.7 | 88.1 | 64.3 | 77.3 | 37.3 |
| MoCo-V3 Chen et al. (2021) | 91.6 | 82.5 | 91.6 | 71.1 | 84.5 | 53.3 | 69.4 | 29.4 |
| SWSL-ResNet50 Yalniz et al. (2019) | 96.1 | 92.0 | 97.4 | 87.8 | 95.1 | 76.3 | 81.7 | 51.2 |
| SWSL-ResNext101-32x16d | 97.2 | 92.7 | 97.9 | 93.6 | 95.9 | 86.3 | 86.7 | 61.0 |

Table 3: Model accuracy on the ImageNet and ObjectNet subsets of our 4,771 images.

| Model | ImageNet | ObjectNet |
|---|---|---|
| ResNet-18 | 79.3 | 47.1 |
| ResNet-18-80% | 74.5 | 44.1 |
| ResNet-18-60% | 74.3 | 42.8 |
| ResNet-18-40% | 70.5 | 39.3 |
| ResNet-18-20% | 63.5 | 31.0 |
| ResNet-50 | 86.6 | 59.3 |
| ResNet-50-80% | 81.7 | 52.2 |
| ResNet-50-60% | 79.6 | 50.5 |
| ResNet-50-40% | 76.3 | 45.1 |
| ResNet-50-20% | 70.4 | 36.9 |
| ResNet-101 | 88.8 | 62.5 |
| ResNet-101-80% | 84.0 | 55.6 |
| ResNet-101-60% | 80.5 | 51.4 |
| ResNet-101-40% | 78.0 | 44.9 |
| ResNet-101-20% | 72.2 | 37.2 |
| ResNet-152 | 89.7 | 64.4 |
| ResNet-152-80% | 83.7 | 54.7 |
| ResNet-152-60% | 81.1 | 51.7 |
| ResNet-152-40% | 77.5 | 46.5 |
| ResNet-152-20% | 71.8 | 37.9 |
| CORNet-S | 80.2 | 51.6 |
| VOneNet-Resnet50 | 83.4 | 53.8 |
| VOneNet-CORNet-S | 80.5 | 47.4 |
| VGG-19 | 80.1 | 49.4 |
| Noisy Student (EfficientNet-L2) | 85.7 | 50.3 |
| DenseNet-121 | 83.7 | 55.2 |
| MSDNet Classifier 0 | 61.7 | 28.9 |
| MSDNet Classifier 1 | 74.0 | 40.9 |
| MSDNet Classifier 2 | 78.3 | 48.3 |
| MSDNet Classifier 3 | 78.3 | 50.6 |
| MSDNet Classifier 4 | 79.4 | 51.0 |
| SimCLR ResNet50 | 75.1 | 45.8 |
| SimCLR ResNet101 | 81.5 | 54.4 |
| SimCLR ResNet152 | 83.4 | 55.7 |
| CLIP-ViT-B/32 | 79.1 | 64.0 |
| CLIP-ViT-B/16 | 84.0 | 74.1 |
| CLIP-ViT-L/14 | 91.8 | 86.0 |
| CLIP-ViT-L/14@336px | 91.6 | 86.7 |
| CLIP-ResNet-50 | 69.8 | 56.5 |
| CLIP-ResNet-101 | 74.5 | 61.0 |
| CLIP-ResNet-50x4 | 76.3 | 64.9 |
| CLIP-ResNet-50x16 | 79.6 | 72.9 |
| CLIP-ResNet-50x64 | 86.6 | 80.3 |
| EfficientNet-S | 82.1 | 47.9 |
| EfficientNet-M | 80.6 | 46.2 |
| EfficientNet-L | 85.5 | 52.2 |
| EfficientNet-S-21 | 89.6 | 65.2 |
| EfficientNet-M-21 | 90.9 | 67.7 |
| EfficientNet-L-21 | 91.4 | 68.8 |
| ViT-T/16 | 58.1 | 28.9 |
| ViT-S/16 | 84.4 | 56.1 |
| ViT-B/16 | 86.6 | 60.7 |
| ViT-L/16 | 94.6 | 77.6 |
| MoCo-V3 | 85.1 | 55.9 |
| SWSL-ResNext101-32x16d | 95.0 | 83.2 |
| SWSL-ResNet50 | 93.5 | 75.3 |
| MAE-ViT-B/16 | 89.6 | 65.0 |

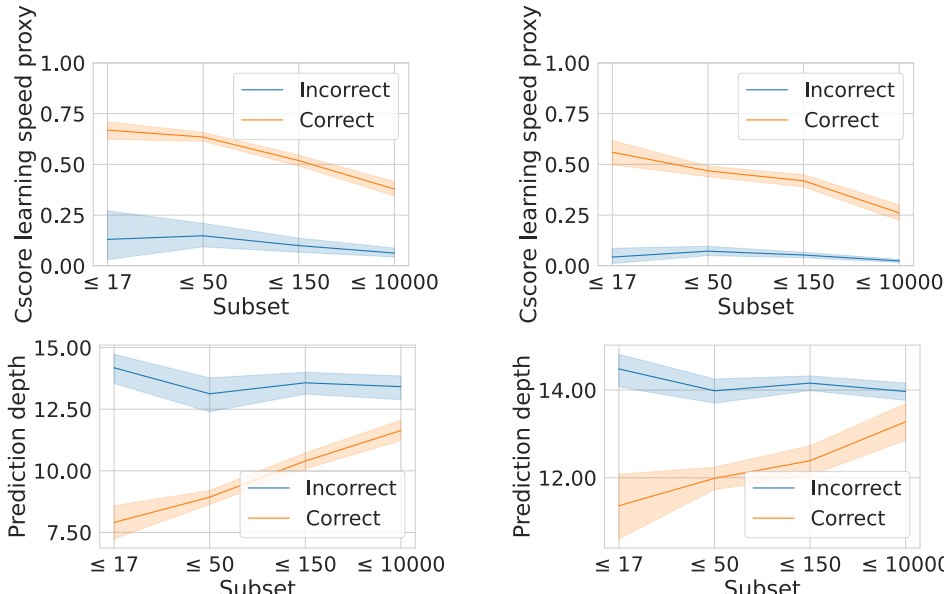

Figure 12: **Top**: left and right are average c-score over subsets for experiment ImageNet and ObjectNet images respectively. Orange shows the images that are correctly predicted by the ResNet-50 while blue shows the images that are incorrectly predicted. **Bottom**: prediction depth plots shown in the same way as top.

### A.6.2 MODEL PERFORMANCE

We evaluate our finetuned models on the same cropped images used in our psychophysics experiments. See table 3 for model accuracy reports on the image difficulty reported in the paper and table 2 for model performance on the full ImageNet and ObjectNet subsets of the experiment images.

### A.7 METRIC CALCULATION PROCEDURE

In this section, we go through the details in computing c-score, prediction depth, and adversarial robustness for our experiment images.

### A.7.1 C-SCORE

C-score (Jiang et al., 2021) identifies individual image difficulty by characterizing the expected accuracy or a held-out image given training sets of varying size sampled from the data distribution. In particular, c-score is the frequency of classifying an example correctly when it is omitted from the training set. However, computing c-score for each image by brute force is computationally infeasible since we must train a separate model for each image. Instead, we computed the learning speed proxy as recommended by the authors. Learning speed measures the epoch at which an image is correctly classified by a model. Intuitively, a training example that is consistent with the training set should be learned quickly because the gradient step for all consistent examples should be similar. The authors found high Spearman rank correlation between c-score and cumulative learning speed based proxies.

We trained a ResNet-50 (He et al., 2016) from scratch on ImageNet1k (Russakovsky et al., 2015) for 90 epochs with an SGD optimizer and initial learning rate of 0.1 with momentum value of 0.9 and weight decay coefficient of 0.0001. The learning rate decayed by a factor of 2 every 9 epochs and the batch size was 256. The standard ImageNet transforms were applied to all images, and the network was initialized randomly. We then evaluated our experiment images at each epoch and used the average of correct predictions as an estimated c-score for each image. fig. 12 shows the average c-scores for ImageNet and ObjectNet experiment images split by whether the ResNet-50 correctly predicted the image. C-score serves as an efficient predictor for human recognition difficulty only for images classified by the model in both ImageNet and ObjectNet. C-scores for images misclassified by

the model do not reveal information about the human recognition difficulty and remain consistently low across all difficulty subsets.

### A.7.2 Prediction Depth

Prediction depth (Baldock et al., 2021) represents the number of hidden layers after which the network's final prediction is already determined. The authors showed that prediction depth is larger for examples that visually appear to be more difficult and is consistent between architectures and random seeds.

We trained a linear decoder at the end of each block of a ResNet-50 on the 50 experiment classes using the ImageNet training and validation set. We used the same ResNet-50 used to calculate c-scores to ensure consistency of our results. There are 16 convolutional layers in a ResNet-50; and each linear decoder follows a convolution layer and consists of a pooling layer, flatten layer, and fully-connected layer. We use the same hyperparameters as appendix A.7.1 and only updated the weights of the linear decoder.

A prediction is defined to be made at depth $L = l$ if the linear classifier after layer $L = l - 1$ is different from the network's final prediction, but the classification of the linear decoder after every layer $L \geq l$ are equal to the final classification of the network. Images classified by all decoders are said to be predicted at layer 0. Note that prediction depth is independent of whether the final prediction is correct or not. It measures the layer at which an image's prediction converges.

Figure 12 shows the average c-scores for ImageNet and ObjectNet experiment images split by whether the ResNet-50 correctly predicted the image. Like c-score, prediction depth serves as an efficient predictor for human recognition difficulty only for images classified by the model in both ImageNet and ObjectNet.

### A.7.3 Adversarial Robustness

We measured an image's distance to the decision boundary of a network using fast gradient sign method (FGSM) (Goodfellow et al., 2015). FGSM creates an modified example that maximizes the loss using the gradients of loss with respect to the input image:

$$mod_x = x + \epsilon \cdot \text{sign}(\nabla_x J(\theta, x, y))$$

where $adv_x$ is the modified image, $x$ is the original image, $y$ is the original input label, $\epsilon$ is a multiplier adjusted accordingly to control the size of modification step, $\theta$ is the model parameters, and $J$ is the loss function. Note that gradients are taken with respect to the input image, and model parameters remain constant.

For an image classified by a model, we define its distance to the closest decision boundary of the model as the minimum $\epsilon$ needed for the model to misclassify the modified image. On the other hand, for an image misclassified by a model, we define its distance to the closest decision boundary of the model as the minimum $\epsilon$ needed for the model to classify the modified image.

We used the same ResNet-50 used to calculate c-scores to ensure consistency of our results. We finetuned the ResNet-50 on the 50 experiment classes using the ImageNet training and validation set. We used the same hyperparameters as appendix A.7.1 and only updated the weights of the final pooling, flatten, and fully-connected layer. We used this finetuned ResNet-50 as the backbone for adversarial perturbation and correction.

While perturbing each classified image, we searched for the smallest $\epsilon$, from 0 to 0.02 incrementing by 1.25e-5 and from 0.02 to 2.5 incrementing by 0.005, that would result in a misclassification. We only applied only one gradient step when perturbing. While correcting each misclassified image, we searched for the smallest $\epsilon$, from 0 to 0.001 incrementing by 1.25e-6 and from 0.001 to 0.05 incrementing by 1.25e-5. We applied two gradient steps when correcting because correction requires finer and more steps.

Note that the search range depends on the backbone model and the dataset. One must choose them through manual trial-and-errors to yield interesting and significant results. Recall that after removing images that were incorrectly annotated, incorrectly cropped, etc section 2.2, we reduced to 4,771 images from the original 5,000. Of these, 3,296 and 1,475 images were classified and misclassified by

the finetuned ResNet-50 respectively. We were not able to find an $\epsilon$ for every image while perturbing and correcting in the corresponding search range. We omitted these images in our analysis. We were able to successfully perturb 2,815 out of 3,296 classified images and correct 1,114 out of 1,475 misclassified images.

We hypothesized that difficult images that are classified and misclassified would be closer and further from the decision boundary respectively. fig. 8 confirms the prior hypothesis. We could not confirm the latter hypothesis due to the smaller number of misclassified images across all subsets, as shown through the higher error bars in fig. 13

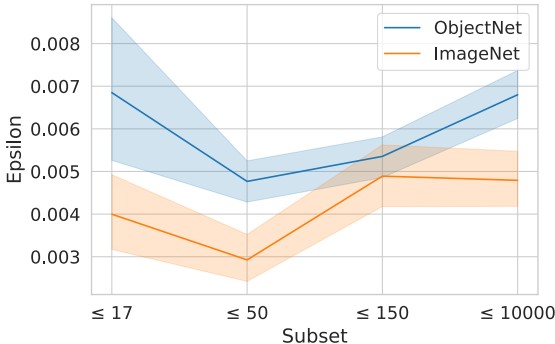

Figure 13: Average $\epsilon$ magnitude required to correct misclassified images back to their correct class per subset

## A.8 CONSTRUCTING A METRIC FOR IMAGE DIFFICULTY

We propose two metrics:

1. Difficulty score which provides an exact ranking from most difficult to recognize to least difficult to recognize based on each response
2. four minimum recognition time (MRT) subsets that quantify the minimum amount of time required for the majority of participants to reliably recognize an image.

Difficulty score is a value from 0 to 28 that represents the number of incorrect predictions given by participants in our experiment across all timings for a particular image. Each image in our experiment was seen an equal number of times per timing and and only rarely were images that were recognizable at shorter timings also recognizable at longer timings. This results in a low difficulty score indicating that an image is easy to recognize and a high difficulty score indicating that an image is hard to recognize. These scores correlate well with the MRT difficulty subsets as shown in fig. 14. Difficulty score varies significantly by object class as well (see fig. 15).

## A.9 DIFFICULTY SCORE DISTRIBUTION BY OBJECT CLASS

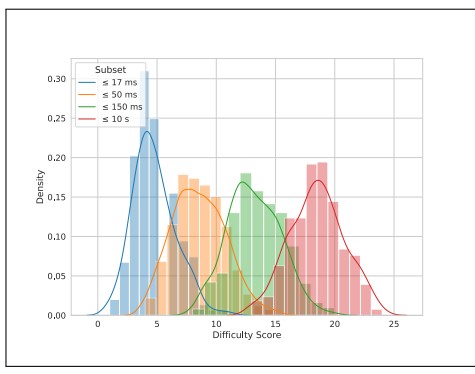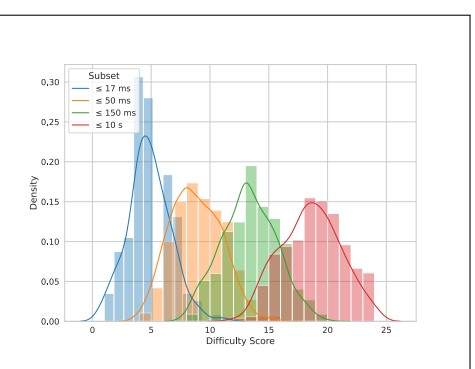

Figure 14: Distribution of difficulty score for each MRT subsets in ImageNet (left) and ObjectNet (right).

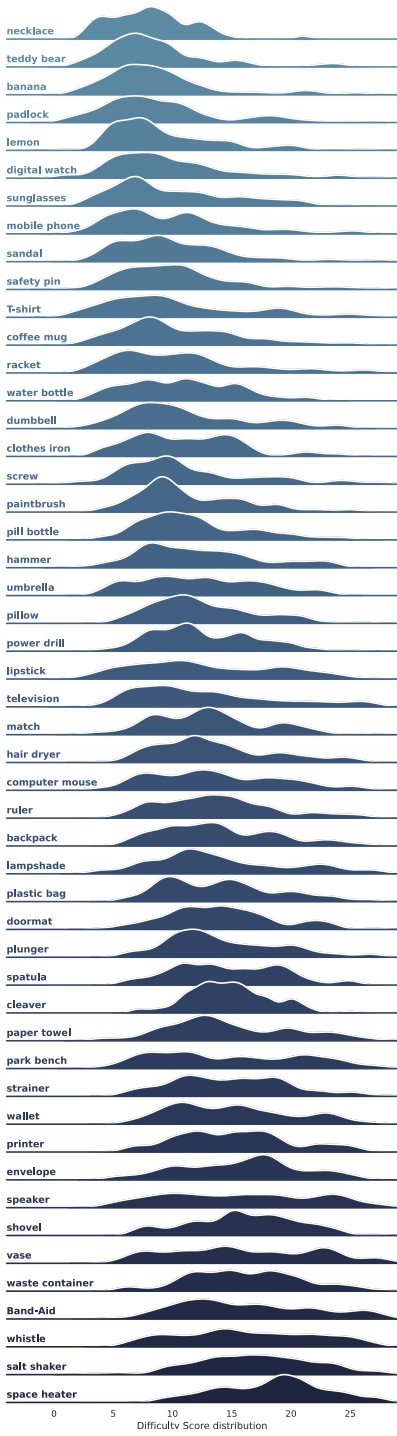

Figure 15: Difficulty distribution by object class sorted in order of increasing mean

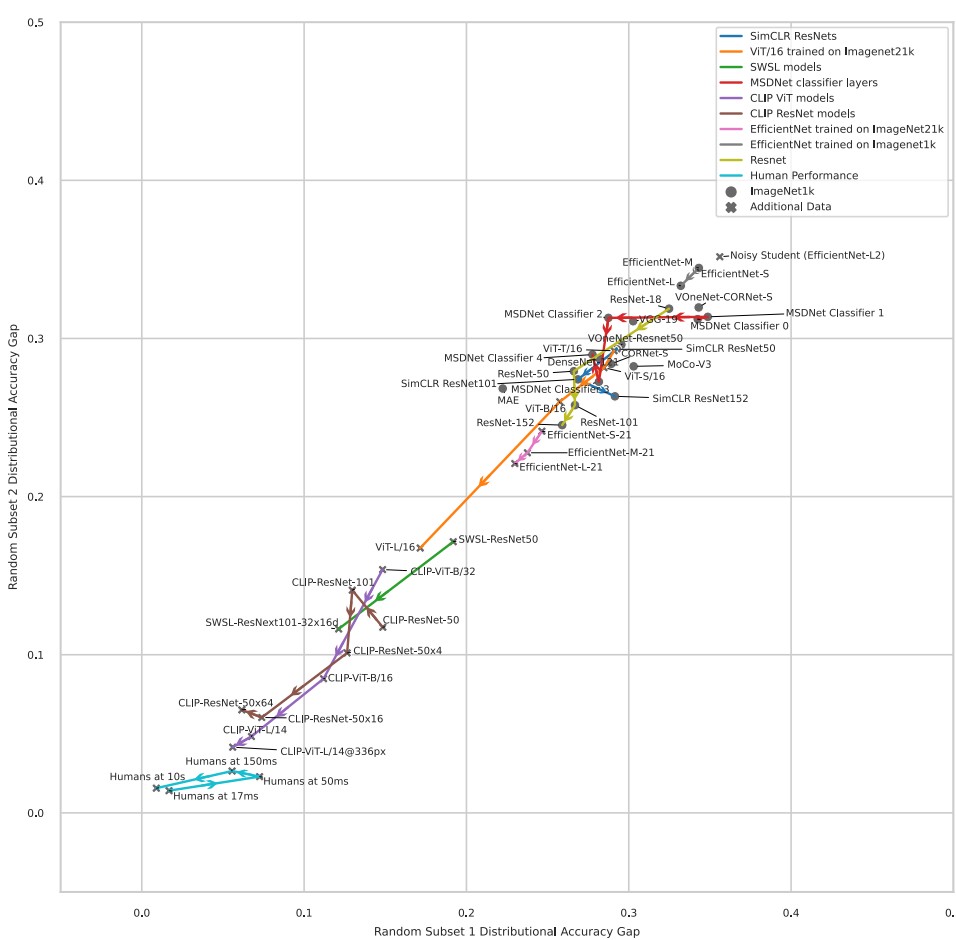

Figure 16: Robustness gap for our finetuned models on two randomly sampled subsets of our experiment data, balanced between ImageNet and ObjectNet. Lines connect model families with arrows pointing in direction of increasing model capacity. Compare with fig. 7.

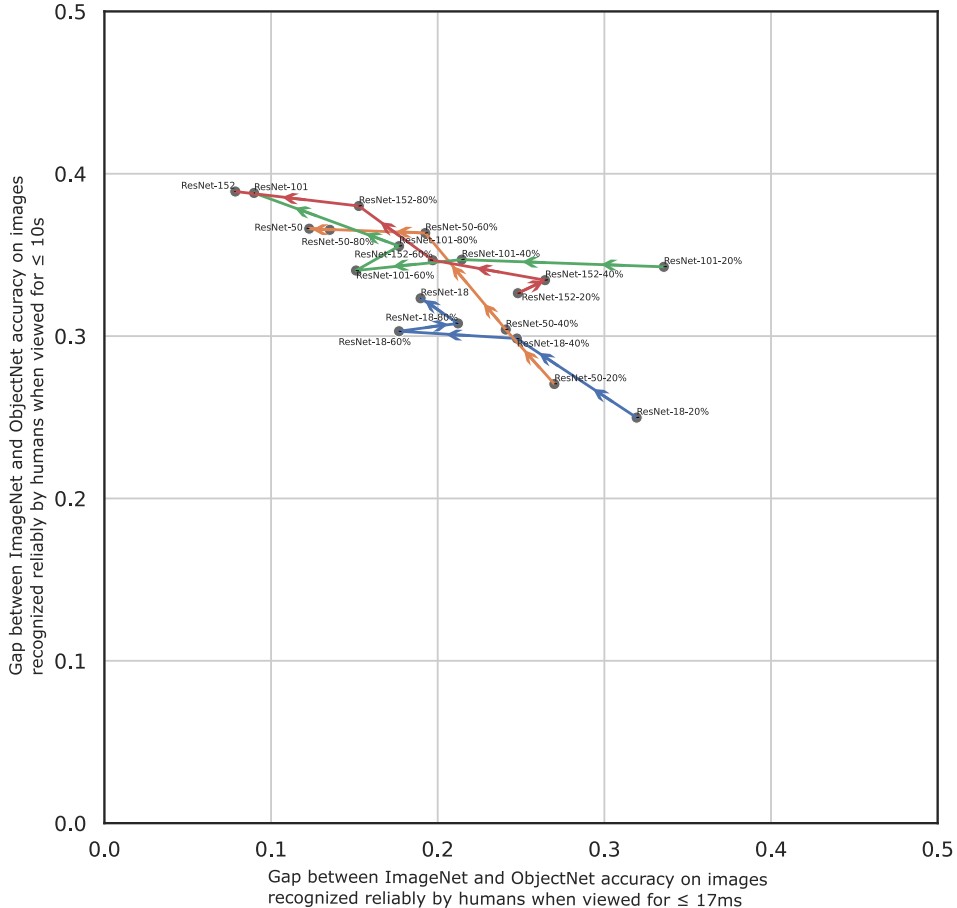

Figure 17: Robustness gap for our finetuned ResNets trained on varying percentages of the ImageNet training set. Lines connect the same architectures with arrows pointing in direction of increasing dataset percentage. Compare with fig. 7.

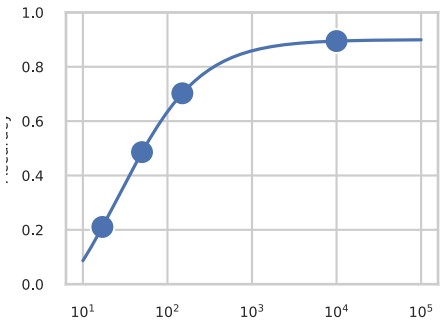

Figure 18: Human accuracy vs Image presentation time from Mechanical Turk results. Time is log-scale with a sigmoid fit

