# OpenReview forum: "How hard are computer vision datasets? Calibrating dataset difficulty to viewing time"
_ICLR.cc/2023/Conference — Submitted to ICLR 2023_

### Official Review · Reviewer_aKwB · 2022-10-24

**Confidence:** 4
**Correctness:** 4
**Technical Novelty And Significance:** 2
**Empirical Novelty And Significance:** 2
**Recommendation:** 5

**Clarity, Quality, Novelty And Reproducibility:**

The paper is very clear and the writing is high quality. The figures are fine, nothing fancy. The work is not very novel, and the results mostly align with Geirhos et al., 2021 who did a similar analysis, but the ability to extract challenge images from any dataset could be important. The results appear to be highly reproducible and I commend the authors for replicating their web-based experiments in the lab.

**Strength And Weaknesses:**


Strengths:
1. Timely investigation of challenging images for humans, with the goal of using those to drive the development of better models.
2. The authors introduce a toolkit for mining existing datasets for these challenging images.
3. A combination of web-based and in-lab experimental validation. Very nice agreement between these experiments.



Weaknesses:

1. Can these images be used to develop better models of AI or human cognition? There's a forward inference/positive result missing here in my opinion. It's not news that CLIP is a good model, so restating that isn't all that interesting to me. It doesn't look like there's any open challenge for AI revealed in these experiments.

2. Does the introduced method identify challenging images consistently, or are some of these images merely poorly labeled exemplars that humans have trouble categorizing?

3. I think the behavioral method is very complicated. Specifically, giving individuals the ability to make a 50-way classification judgement. We don't know the psychological distances between those categories so it's difficult to say whether or not a preponderance of certain categorical choices could makes it easier/harder to find a given category (i.e., it might be harder to distinguish between species of dogs )

4. Make sure you ref [1] as that is most similar to this work for OOD classification.

5. Measuring difficulty in humans is tough -- I agree that there should be images that are unrecognizable in fast presentations, but these same images should be recognizable with enough time. Otherwise, I am worried that the objects are just mislabeled. What do you think?

6. Figure 7 is really cool but also complicated. Would it be possible to add some additional plots to highlight the insights you describe in the text about SimCLR etc?

7. "We are of course not the first to carry out such viewing time experiments" I think you should cite the much longer history of rapid visual classification experiments: [2], [3], [4] to give a sampling of works over the years that investigate performance on natural image databases (ImageNet in the case of [4]) as a function of stimulus exposure time. The goal of these works is to limit viewing time to better interogate visual system mechanisms associated with the "feedforward" sweep.

8. I suspect the findings of c-score and Figure 8 could be idiosyncratic for different architectures. For instance, "This analysis reveals that images that require more viewing time... are predicted by later layers in the network." This is consistent with the plethora of studies analogizing ResNets and RNNs, which say that the skip connections enable stronger non-linearities and as a result the ability to process more complicated stimuli. What happens when you try other architectures like the ViT in CLIP?

[1] Geirhos et al. Partial success in closing the gap between human and machine vision. 2021.

[2] Fabre-Thorpe. The Characteristics and Limits of Rapid Visual Categorization. 2011.
[3] Serre et al. A feedforward architecture accounts for rapid categorization. 2007.
[4] Eberhardt et al. How Deep is the Feature Analysis underlying Rapid Visual Categorization? 2016.


**Summary Of The Paper:**

The authors introduce a behavioral approach for identifying challenging images in large image datasets. They use their method to find thousands of challenging images in ImageNet and ObjectNet, and find that CLIP has to a great extent matched humans on these images. Through further analyses they elaborate on the failures seen by different classes of models. They release their datasets and code for the community to build off.

**Summary Of The Review:**

I am totally borderline on this. I think the method for curating datasets is important but I have some quibbles about its details (50-way classification potentially introduces issues in the data — totally non-ecological). I think some of the analyses are interesting but there's no key finding here I can take to my group to say this is changing the way we think about AI or human vision. I look forward to seeing what the other reviewers say. The work is high quality but the impact of the presented results is what keeps me from recommending acceptance.

---

> ### Author Response · Authors · 2022-11-18
> **Review response (1/2)**
>
> > Can these images be used to develop better models of AI or human cognition? There's a forward inference/positive result missing here in my opinion. It's not news that CLIP is a good model, so restating that isn't all that interesting to me. It doesn't look like there's any open challenge for AI revealed in these experiments.
>
> The reviewer is right that previous work pointed out that CLIP is better; but this is not our contribution. Previous work also shows that other models are quite similar in performance to CLIP, such as SimCLR-trained models. We show something entirely different. That CLIP and other models follow fundamentally different scaling laws. Scaling up other models leads to better performance, but only closes out of distribution performance gaps on the easy images. CLIP closes performance gaps on both.
>
> This observation about scaling laws on different sets of images (what happens as CLIP and other models get bigger) is novel and important. It says there is something fundamentally different about CLIP, it is not a matter of just raw performance.
>
> > Does the introduced method identify challenging images consistently, or are some of these images merely poorly labeled exemplars that humans have trouble categorizing?
>
> We worried about this as well and made sure to address it during the experiment. Humans have no difficulty labeling these images. With 10 seconds of viewing time, people have nearly 100% performance. That shows the labels are correct, 7 independent people agree that the correct object is shown on a 1-out-of-50 task (2% chance). While running the experiment, we eliminated all images, 229, that humans could not agree on with 10 seconds of viewing time. This process is briefly discussed in Section A.5.
>
> > I think the behavioral method is very complicated. Specifically, giving individuals the ability to make a 50-way classification judgement. We don't know the psychological distances between those categories so it's difficult to say whether or not a preponderance of certain categorical choices could makes it easier/harder to find a given category (i.e., it might be harder to distinguish between species of dogs)
>
> We understand, and coincidentally, our original goal was to measure the relative distances between categories! But then we realized that image-level difficulty metrics are very important and lead to revelations about model performance and datasets.
>
> Note that humans had no difficulty with this experiment, they achieved nearly 100% performance with 10 seconds of viewing time. The experiment requires some care of course, and a 1-out-of-50 discrimination task is challenging, but well within the capabilities of Mechanical Turk workers.
>
> We agree with the reviewer. Some classes are easier than others. What might be an easy image of one object class would be a very challenging image of another object class. And one could use such judgements to establish the relative difficulty of different object class discrimination tasks (that class A is very hard to distinguish from B; as with the reviewer’s example of dogs). Indeed, looking at the error patterns in the data, we could go down this route and investigate which classes are most confusable.
>
> That being said, this doesn’t change the bottom-line story. Datasets have an underlying image difficulty distribution and we are currently ignoring it. It’s true that there are many factors that can lead to that distribution, one of which is object class. We are excited to follow up on these issues of class difficulty.
>
> > Make sure you ref [1] as that is most similar to this work for OOD classification.
>
> Thanks! We will add it.
>
> > Measuring difficulty in humans is tough -- I agree that there should be images that are unrecognizable in fast presentations, but these same images should be recognizable with enough time. Otherwise, I am worried that the objects are just mislabeled. What do you think?
>
> We agree! With 10 seconds of viewing time, humans have 100% agreement on the 1-out-of-50 task (2% chance). This means they could recognize those images, so any failures are due to time constraints. We were very worried about this exact problem while designing the experiment and controlled for it. We removed images that humans couldn’t label even with 10 seconds of viewing time; those images are just mislabeled. The details of this process are described in A.5
>
> > Figure 7 is really cool but also complicated. Would it be possible to add some additional plots to highlight the insights you describe in the text about SimCLR etc?
> Thanks! We will consider other plots that we could add to the appendix to more clearly demonstrate these model family scaling trends. We have also included a more complete table of models performance on different difficulty subsets across ObjectNet and ImageNet in the appendix.

---

> > ### Author Response · Authors · 2022-11-18
> > **Review response (2/2)**
> >
> > > "We are of course not the first to carry out such viewing time experiments" I think you should cite the much longer history of rapid visual classification experiments: [2], [3], [4] to give a sampling of works over the years that investigate performance on natural image databases (ImageNet in the case of [4]) as a function of stimulus exposure time. The goal of these works is to limit viewing time to better interogate visual system mechanisms associated with the "feedforward" sweep.
> >
> > Thank you. We will do so!
> >
> > > I suspect the findings of c-score and Figure 8 could be idiosyncratic for different architectures. For instance, "This analysis reveals that images that require more viewing time... are predicted by later layers in the network." This is consistent with the plethora of studies analogizing ResNets and RNNs, which say that the skip connections enable stronger non-linearities and as a result the ability to process more complicated stimuli. What happens when you try other architectures like the ViT in CLIP?
> >
> > Good question! We know it’s idiosyncratic for datasets. For example, we computed c-score proxies for CIFAR10 and CIFAR100 and discovered that the same images found in both had much higher c-scores (i.e. easier to learn) in CIFAR10 than in CIFAR100 due to the larger number of classes and increased difficulty of learning the training distribution. An objective image difficulty metric should be agnostic to the image distribution. These correlations are likely also idiosyncratic for models. These idiosyncrasies actually served as a motivation for developing more consistent difficulty metrics that are not dependent on datasets or models. We would love the answer to the reviewer’s question, unfortunately, it isn’t feasible. C-score is computationally very expensive, well beyond our means or that of most labs, as it requires retraining ViT or CLIP under different conditions. Even the proxies for c-score are prohibitively expensive for such models. The one metric that is computable, prediction depth, shows nothing unremarkable about ViT or CLIP, we will update the manuscript to include it.
> >
> > > The paper is very clear and the writing is high quality. The figures are fine, nothing fancy. The work is not very novel, and the results mostly align with Geirhos et al., 2021 who did a similar analysis, but the ability to extract challenge images from any dataset could be important. The results appear to be highly reproducible and I commend the authors for replicating their web-based experiments in the lab.
> >
> > Thanks for the kind words! We draw very different conclusions from Geirhos et al. 2021; our work is mostly orthogonal, and the impact of our work is very different.
> >
> > 1. Our results show that Geirhos et al. 2021 put humans at a significant disadvantage. They limit humans to 200ms of viewing time, which lowers their performance by 20% or so in current datasets. They speculate about the impact of this decision, but cannot quantify it.
> > 2. Geirhos et al. 2021 do not investigate dataset difficulty at all. Indeed, all of their results are confounded by difficulty, since they use images sampled from existing datasets. This means that overwhelmingly they are testing the gap between humans and machines on easy images, not hard images.
> > 3. Image difficulty in essence explains some of the results by Geirhos et al. 2021. The gap between humans and machines is small on easy images, but large on hard images.
> > 4. Geirhos et al. 2021 cannot be used to guide new dataset creation. Here, we provide a practical path forward in two ways. That datasets should report their difficulty curves explicitly so that we can compare them more objectively. And that dataset creation can monitor this curve and explicitly tune it. Moreover, models could report performance as a function of image difficulty.
> > 5. We show a new scaling law that doesn’t appear in that work. That CLIP follows a different path when scaling up compared to other models.
> >
> > This is not an exhaustive list and we hope the reviewer will not take our reply as being negative about Geirhos et al. 2021. We believe that psychophysics has much more to contribute to computer vision.
> >
> > We believe the impact of understanding dataset difficulty can be significant. From how datasets are developed, to how they are reported, to which new datasets we adopt – difficulty measures can guide us in an objective way in what was otherwise a largely subjective enterprise. In addition to helping guide model development by isolating images that models have difficulty with but humans do not; or equivalently by reporting model performance as a function of dataset/image difficulty.

---

### Official Review · Reviewer_ZZZd · 2022-10-27

**Confidence:** 4
**Correctness:** 3
**Technical Novelty And Significance:** 3
**Empirical Novelty And Significance:** 4
**Recommendation:** 8

**Clarity, Quality, Novelty And Reproducibility:**

This is a very well written and high quality work. The idea itself may not be original, but the way the problem was studied and analyzed is novel. Dataset and analysis code is released publicly.

**Strength And Weaknesses:**

Strength

* Paper is very well written and easy to follow. Experiments are well designed and systematically executed. Analyses and questions asked in the paper are thought provoking and high quality.
* Demonstrated the feasibility of this metric as a difficulty measure and identified key next steps we need to work towards as a community.
* The proposed approach is much more scalable than gaze tracking and can be potentially adopted by others with low cost.


Weakness

* Further discussion on what would be a good way to incorporate this metric intro a new data collection protocol  would be useful; how can researchers plan ahead and target at collecting more harder samples in the beginning?
* It remains unclear whether viewing time would generalize well and serve as a good difficulty metric for other vision tasks such as optical flow


I have a minor question on the study design; a set of words are displayed at the end of the viewing session - what if the viewer is not familiar with certain word and fails to answer it correctly because they do not know how the object looks like, but not because they do not recognize the image?


**Summary Of The Paper:**

This paper studies a new way of measuring difficulty of computer vision datasets. The authors propose to use humans’ viewing time as a difficulty measure and show that longer viewing time correlates with difficulty. They also show that the measured difficulty can be computationally modeled with a combination of prediction depth, c-score, and adversarial robustness. Analysis of SOTA models reveals that there exists larger gap on harder images.

**Summary Of The Review:**

This paper studies an important problem for the computer vision community. The experiments and analysis are well organized and provides new insights. Given the overall strengths of the paper, I recommend acceptance.

---

> ### Author Response · Authors · 2022-11-18
> **Review response**
>
> > Further discussion on what would be a good way to incorporate this metric intro a new data collection protocol would be useful; how can researchers plan ahead and target at collecting more harder samples in the beginning?
>
> Good question! As a dataset is being collected, researchers can use our tools to look at the distribution of their dataset. Then, look at the hard and easy images themselves. This will quickly reveal the kinds of conditions in that domain which one should focus on. As the dataset is being gathered, this process can be repeated (the low cost and low effort to measure difficulty with our automated tooling pales in comparison to the effort involved in gathering a new dataset), and researchers can have objective feedback if their efforts to change the dataset difficulty are helping. Moreover, they could then report/understand the performance of models as a function of difficulty.
>
> > It remains unclear whether viewing time would generalize well and serve as a good difficulty metric for other vision tasks such as optical flow
>
> As long as visual search is not a factor, many other vision tasks can be addressed. If you mean, optical flow over a several-second video, then no. But, if we mean, dense optical flow on nearly-adjacent frames while focused in on one region? Then yes, we actually give this as an example of where we want to move next. For example, instead of showing 1 image for 17ms a sequence of two images could be shown for 17ms each followed by a question about the motion between the two images. Other tasks such as segmentation, colorization, depth estimation, etc. can also be investigated.
>
> Good question! Participants were shown the list of image words prior to beginning the experiment. They were told that they do not need to memorize the words, but that they should look at them now and make sure they know all the words. They were encouraged to look up pictures on the internet if they were unfamiliar with one of the words. Some participants in the in-lab experiment were not native English speakers and took advantage of this preparation page to look up images of the words they did not know. The in-lab results show no difference between native English speakers and second-language speakers. Although we can’t confirm that all of the MTurk participants did their due diligence to familiarize themselves with the classes, the consistency of the data with the in-lab experiment gives us confidence in their responses. Additionally, the 50 object classes that we chose to include in this experiment are all common household objects that we expect most participants were already familiar with.
>
> > This paper studies an important problem for the computer vision community. The experiments and analysis are well organized and provides new insights. Given the overall strengths of the paper, I recommend acceptance.
>
> Thank you for your thoughtful review!

---

### Official Review · Reviewer_y1Qy · 2022-11-03

**Confidence:** 3
**Correctness:** 3
**Technical Novelty And Significance:** 3
**Empirical Novelty And Significance:** 2
**Recommendation:** 5

**Clarity, Quality, Novelty And Reproducibility:**

The paper is relatively well written and easy to follow. The main idea of the paper is clear and well presented. The practical benefit of the difficulty function and the main claim though is somewhat questionable.

**Strength And Weaknesses:**

Strength:
- Novel metric for evaluating image difficulty.
- Valuable and deep analysis of the difficulty, easy and hard samples of ImageNet and ObjectNet datasets.
- The analysis of various model performance on the samples of different difficulty
- Promising results that can help constructing the new datasets for various Computer Vision Tasks

Weaknesses:
- The process of determining the difficulty of the sample is manual and tedious. The authors did not suggest the way to automate the process for the other datasets. It would be interesting to experiment whether a neural network can learn the sample difficulty function from the labeled data and whether this can be used to automatically select samples for labeling
- The correlation of the model performance and the sample difficulty is expected but it is not proved whether additional "hard" samples would improve the model performance. As shown in the paper many of the "hard" samples do contain various degradations (occlusions, blur, bad illumination, etc.) The authors might perform a test training a model on 2 versions of the dataset: with less and more "hard" samples and evaluating the performance on a fair large scale test dataset.
- The work does not introduce the updated version of the dataset and does not elaborate what the optimal distribution of the sample difficulty should be.
- Some of the plots (Figure 7) are hard to follow, authors can change the format or the quantity of the models on the plot to improve the readability.

**Summary Of The Paper:**

The paper introduce a novel dataset difficulty metric based on how long humans have to view an image in order to classify it correctly. The authors release the difficulty metrics for ImageNet and ObjectNet datasets as well as distribution of image difficulties in those datasets. The paper also introduce a new metric predicting object difficulty.

**Summary Of The Review:**

The paper address the important topic of evaluating the dataset quality and difficulty and mining the hard samples. The authors introduce the way to manually compute the difficulty of the sample based on how ling it takes for a human to classify it. Since the process can not be automated for other tasks and it is not fully proven that more hard samples by introduced metric lead the the better model performance the benefit of the suggested metric is debatable. The additional experiments can prove the main concept and increase the usability of the suggested solution

---

> ### Author Response · Authors · 2022-11-18
> **Review response (1/2)**
>
> > The process of determining the difficulty of the sample is manual and tedious. The authors did not suggest the way to automate the process for the other datasets. It would be interesting to experiment whether a neural network can learn the sample difficulty function from the labeled data and whether this can be used to automatically select samples for labeling
>
> Compared to current metrics that are a function call away, yes, this is more involved. But, it is an almost entirely automated process. Currently, the only steps involved are automatically cropping out bounding boxes, running a script to submit tasks to Mechanical Turk, and then interacting with the MTuk interface. We will release our tool that automates this process. We are in the process of moving the tool to Prolific and will hopefully have a completely automated solution where the only difficulty is providing a few hundred dollars.
>
> In the manuscript, we address the question of predicting difficulty by calling on previous work that has attempted to measure image difficulty. Popular model-based measures of sample difficulty that exist now (c-score, prediction depth) correlate with our metric but do not capture all the variation in the data. Additionally, these existing metrics are not objective image-level metrics, i.e., they rely on trained models and are biased by those models. This leads to wild mispredictions and failures out of distribution, which is exactly when these tools are needed.
>
> > The correlation of the model performance and the sample difficulty is expected but it is not proved whether additional "hard" samples would improve the model performance. As shown in the paper many of the "hard" samples do contain various degradations (occlusions, blur, bad illumination, etc.) The authors might perform a test training a model on 2 versions of the dataset: with less and more "hard" samples and evaluating the performance on a fair large scale test dataset.
>
> This is an interesting point! There are two separate questions behind it. One is, how do you evaluate models in a way that gives you confidence they will perform well in the real world? The other is how do we train the models to be more accurate? We focused on the first here. Many deployments of computer vision are coming, and understanding their potential failures is critical. Moreover, harder datasets open up the doors to creating better models; it is hard to imagine how we will find new ideas and architectures on datasets that current models mostly solve.
>
> The training question is of course interesting. Can we use hard images to improve models? We note that this requires the first question to be addressed: you must first have a dataset that is hard enough, where this question matters at all! We did not address this question because we don’t have enough data to do it justice. We are thinking about how to do this in a follow-up publication.
>
> > The work does not introduce the updated version of the dataset and does not elaborate what the optimal distribution of the sample difficulty should be.
>
> While we didn’t give the dataset an explicit name, we do release such a dataset and it is available now. That’s a fair note; we will think of a name and add it to the manuscript.
> The human experiment data and the code necessary to reproduce all our results can be found here: https://github.com/image-flash/image-flash-2022 (link included in abstract). The images from ImageNet and ObjectNet need to be downloaded separately, but our data release includes code that will crop the images as they were cropped in the experiment (see appendix), and that will identify how images were characterized in our experiments according to minimum recognition time.
>
> We don’t think there is an optimal sample difficulty. Instead, we think that systems should report performance as a function of difficulty. Of course, this means that datasets should strive to contain enough hard examples to enable this. And that we should acknowledge that performance averaged across a dataset that is highly skewed, i.e., mostly easy, is essentially only reporting performance on the easy images. With the graph of performance vs difficulty in front of them, users of models can decide what matters to them most. In the process, we will also have a more refined idea of what performance means. And a better target for neuroscience and behavioral experiments.
>
> > Some of the plots (Figure 7) are hard to follow, authors can change the format or the quantity of the models on the plot to improve the readability.
>
> Very fair point. We received similar feedback from other reviewers and will brainstorm how to make the point of the figure more clear.

---

> > ### Author Response · Authors · 2022-11-18
> > **Review response (2/2)**
> >
> > > Summary Of The Review: The paper address the important topic of evaluating the dataset quality and difficulty and mining the hard samples. The authors introduce the way to manually compute the difficulty of the sample based on how ling it takes for a human to classify it. Since the process can not be automated for other tasks and it is not fully proven that more hard samples by introduced metric lead the the better model performance the benefit of the suggested metric is debatable. The additional experiments can prove the main concept and increase the usability of the suggested solution
> >
> > Thank you for your thoughtful review.
> >
> > We note that the process is automated. We have released and will continue to refine tooling that given images and bounding boxes and several hundred dollars in a Mechanical Turk or (soon) Prolific account, computes these metrics. We are continuing to make it more automatic.
> >
> > We do not claim that additional hard examples make models better. It would be good if this was the case! And, it might be. Our focus here is on testing models in a more effective way.
> >
> > Right now, because of how skewed datasets are, the performance numbers we compute are essentially the performance over the easy images. There are very few hard images and so lower performance over those images has a small impact on the overall average performance.
> >
> > There is great interest in computer vision, and other fields, on model failures now that average performance is quite high. This is largely being done by looking at distribution shift to new datasets. Here we point out that this isn’t enough. ObjectNet was designed specifically to address issues of distribution shift. But ObjectNet is no more difficult than ImageNet for humans. This means that we should focus, in addition to distribution shift, on an orthogonal quantity: hardness.
> >
> > By ensuring that we even out the difficulty of datasets to include images that are hard and report performance as a function of image difficulty, we will evaluate models in a more refined way. This means that we can understand, predict, and explain many failures in a way we could not before: the model failed on these images because they’re hard for humans and models have difficulty with such images. This is invaluable for downstream applications, both in terms of understanding what will and won’t work, and in terms of knowing what to expect from models.
> >
> > It is also important for improving performance. It may well be, as the reviewer suggested, that performance will go up by training on such images. But, there is a different view of performance, one that applies to our community as a whole. We design models, their architectures and hyperparameters, to our datasets. Datasets that are too easy lead us as a community to overfitting the hyperparameters of our models to such easy images. We believe this is where difficulty can have the biggest impact: by demonstrating a gap in current capabilities, it can provide a better target for models that are currently only able to show incremental improvements over existing datasets.
> > We are of course also very interested in understanding how hard images can improve models directly. But the mere fact that we can identify a subset of images in an objective manner that is overlooked by current datasets, that is pervasive in the vision community, that human vision solves, and that models have difficulty with, is, we believe, very valuable.

---

### Official Review · Reviewer_KToy · 2022-11-03

**Confidence:** 3
**Correctness:** 4
**Technical Novelty And Significance:** 3
**Empirical Novelty And Significance:** 3
**Recommendation:** 6

**Clarity, Quality, Novelty And Reproducibility:**

While the writing could be shored up in some places (e.g. instances of premature sentence
full-stops and abrupt transitions) the paper is generally well-written, with the premise,
methodology, and analysis all clearly and succinctly presented. The figures are generally germane and
well-constructed, though the grouping by presentation time along two axes in Figure 5 is slightly
confusing. Details of the experimental procedures, both psychophysical and computational, are given
in the main text and expanded upon in the appendices. While on the surface similar to works such as
Geirhos et al. 2021 [1], the paper adopts similar procedures to pursue the complementary objective
of evaluating the distribution of sample difficulty within two prominent object-recognition datasets.

[1] Geirhos R, Narayanappa K, Mitzkus B, Thieringer T, Bethge M, Wichmann FA, Brendel W. Partial
success in closing the gap between human and machine vision. Advances in Neural Information
Processing Systems. 2021 Dec 6;34:23885-99.


**Strength And Weaknesses:**

### Strengths
- The motivation and its distinction from prior work is clearly established -- while similar
  undertakings have been conducted before, the paper does explore a novel avenue in attempting to
  quantify sample difficulty in a way decoupled from any given model via psychophysical trials.
- The scale of the psychophysical experiments is commendable as is the rigor invested in trying to
control them and validate those conducted under less-controlled conditions (Mechanical Turk).
  While the results for easier samples may not be entirely admissible, for the hard samples that
  are of primary interest the two groups do seem to align reasonably well.
- Figures are mostly clear and germane, with descriptive captions, though some figures are not
obvious without reading of said captions (namely, Figure 5 and 7).
- Discussing limitations in the experimental procedure is not shied away from -- the authors, for
  instance, identify and concede issues with the shorter presentation times, stemming from
  the use of crowdsourcing.
- The authors are able to convincingly show correlation between their psychophysics-derived
  metric and the performance of neural networks across a wide range of architectures. They make a
  sound case, consistent with other recent literature, that current benchmark datasets for object
  recognition are lacking in their ability to capture real-world diversity.
- The authors show with large-scale experimentation that viewing time can serve as a valid
metric for sample difficulty across a variety of architectures.
- The appendices are in their detailing of experimental procedure and materials and in presentation
  of additional results.

### Weaknesses
- There is some important related work missing. Most notably, while the authors cite Geirhos et al.
2018, they neglect to follow up the follow-up work from 2021 [1] which extends the suite of psychophysical
trials and shows a closing gap in the OOD performance of humans and machines. Although I understand
the goals of said work and the present work to be complementary (the former aiming to evaluate the
differences in robustness, the latter aiming to devise an 'objective' measurement for sample
difficulty), given that the works employ similar methodologies to establish a benchmark with which
to contrast human vs. machine performance (albeit under different regimes), I think some comparison
is warranted.
- As alluded to in the above section, while I understand the rationale behind the design choices
for Figure 7, it's not an easy figure to navigate visually. Similarly, Figure 5 is confusing for
its use of presentation time both as a categoriser and as an axis (though I realise an
explanation of this is given in the accompanying caption).
- The proposed method for evaluating sample difficulty is only obviously applicable to datasets of the same
type as ImageNet, that is those that are focused on single objects -- there's no obvious way to extend
the procedure to instance segmentation datasets, for instance, in which many objects may be
present in a scene and the task goes beyond simple image-level recognition.
- Much of the introduction reads rather informally rather than as academic prose.

(very minor) gripe: The reference associated with Geirhos et al. 2018 is an older (arxiv) version of the
text, not the version published at NeurIPS in 2018

[1] Geirhos R, Narayanappa K, Mitzkus B, Thieringer T, Bethge M, Wichmann FA, Brendel W. Partial
success in closing the gap between human and machine vision. Advances in Neural Information
Processing Systems. 2021 Dec 6;34:23885-99.


**Summary Of The Paper:**

The authors propose to conduct psychophysical experiments using a combination of in-lab and
crowdsourced subjects to derive an 'objective' (that is, one decoupled from any particular model)
measure of the distribution of sample-difficulty in two widely-used object recognition datasets,
ImageNet and ObjectNet. This differs from recent work that conducts similar experiments to assay
the gap between existing models and humans on OOD benchmarks and derive a benchmark using the
collected traits. The authors propose to use 'minimum viewing time needed to recognise an object
within an image' as a proxy for sample difficulty, with an object considered 'recognised' if more
than half of participants could classify it. Four presentation times are considered for
categorisation, with objects recognisable within 17ms constituting the easiest samples, objects
recognisable within 10s constituting the hardest samples. The authors find that ImageNet and
ObjectNet are greatly skewed towards easier samples according to the proposed metric and suggest
increasing a representation of harder samples to be an important avenue for future work on creating
more robust models. The crowdsourced data is compared the data obtained under controlled lab
conditions and found to be reasonably-well aligned, with discrepancies mainly occurring at the
shorter presentation times. The authors show that the proposed measure of difficulty correlates
with performance-degradation in a variety of models and furthermore that this measure can be
predicted to some degree using model-based difficulty measures such as c-score and prediction
depth.


**Summary Of The Review:**

The paper conducts an impressive array of both psychophysical -- involving human subjects -- and
computational experiments in order to derive an validate an objective measure of sample-difficulty
for ImageNet and ObjectNet. While I'm not convinced about the extensibility of the method
to problems beyond ImageNet-style datasets, the paper addresses and attempts to quantify an
increasingly-apparent problem; proposes and empirically justifies, through comprehensive analysis, an
intuitive proxy metric for the datasets in question (which are among the most-used benchmarks in
ML); provides interesting insights into the datasets in question using the derived metric; is generally
well-written (consistently clear and concise); and does well in outlining the procedures for the psychophysical trials
and addressing the ethics and limitations involved.

---

> ### Author Response · Authors · 2022-11-18
> **Review response**
>
> > There is some important related work missing. Most notably, while the authors cite Geirhos et al. 2018, they neglect to follow up the follow-up work from 2021 [1] which extends the suite of psychophysical trials and shows a closing gap in the OOD performance of humans and machines.
>
> Yes, we should have cited this paper and will do so. We are very aware of the complementary nature of recent work from Geirhos et al. and will make sure to include a comparison in our revision.
>
> > As alluded to in the above section, while I understand the rationale behind the design choices for Figure 7, it's not an easy figure to navigate visually. Similarly, Figure 5 is confusing for its use of presentation time both as a categoriser and as an axis (though I realise an explanation of this is given in the accompanying caption).
>
> Thanks for the feedback! We had the same concerns, but both figures make important points. We have thought at length about how to better present the same data. Do you have any suggestions?
>
> > The proposed method for evaluating sample difficulty is only obviously applicable to datasets of the same type as ImageNet, that is those that are focused on single objects -- there's no obvious way to extend the procedure to instance segmentation datasets, for instance, in which many objects may be present in a scene and the task goes beyond simple image-level recognition.
>
> The methodology is much more generic than just object recognition. For example, depth estimation, edge detection, optical flow estimation, object segmentation, reconstructing occluded parts of objects, colorization, and more. Many computer vision tasks can be performed either at the individual object level or the image-patch level.
> A concrete example of another application of this methodology could be in medicine. How hard are current tumor segmentation datasets? Are they mostly fairly easy cases? This could be tested image patch by image patch.
> That being said, yes, there are tasks that are out of scope for this method. Tasks that have to do with locating objects are incompatible. But this is a minority of tasks in computer vision.
>
> > Much of the introduction reads rather informally rather than as academic prose.
>
> Thanks for the feedback! We’ll clean up the language so it reads better.
>
> > Summary Of The Review:
> The paper conducts an impressive array of both psychophysical -- involving human subjects -- and computational experiments in order to derive an validate an objective measure of sample-difficulty for ImageNet and ObjectNet. While I'm not convinced about the extensibility of the method to problems beyond ImageNet-style datasets, the paper addresses and attempts to quantify an increasingly-apparent problem; proposes and empirically justifies, through comprehensive analysis, an intuitive proxy metric for the datasets in question (which are among the most-used benchmarks in ML); provides interesting insights into the datasets in question using the derived metric; is generally well-written (consistently clear and concise); and does well in outlining the procedures for the psychophysical trials and addressing the ethics and limitations involved.
>
> Thanks for the thorough and thoughtful review! We appreciate your assessment of the value of our contribution in the object recognition task.
>
> We believe that many if not most computer vision tasks are compatible with this approach. Of course, not all are. For example, object segmentation can be tested in this way, optical flow, colorization, object part identification, etc. Essentially, tasks that don’t require visual search. Other tasks can be adapted to this paradigm. For example, even visual question answering can be adapted, factoring out the search part; you would identify which part of the image the question is about and then present only those parts.
>
> At a higher level, we believe that the community can gain insights into computer vision tasks, models and datasets, through the lens of human difficulty. Image classification is a natural place to start because of its popularity and high impact. We look forward to applying this idea to other areas.

---

### Decision · Program_Chairs · 2023-01-20

**Decision:**

Reject

**Justification For Why Not Higher Score:**

N/A

**Justification For Why Not Lower Score:**

N/A

**Metareview: Summary, Strengths And Weaknesses:**

This paper makes an attempt towards constructing more challenging image datasets inspired by cognitive studies. This work proposes to measure dataset difficulty based on how long humans have to view an image to classify a target object. Image difficulties collected in two large datasets, ImageNet and ObjectNet, are among the main contributions of the work.

The paper got mixed scores from accept, to borderline reject, remaining borderline overall.
While acknowledging that the proposed metric of dataset difficulty is potentially useful, the reviewers have provided detailed comments and raised concerns. Most critical concerns are related to 1) generalisation to other datasets/tasks; this is important to better assess the scope and significance of contributions (all reviewers), 2) lack of significant/unexpected findings w.r.t. prior work [Geirhos et al 2021], and 3) writing and presentation clarity of the paper.
Based on our discussion about the work, it was concluded that the paper has contributions worth publishing, but it will be most appropriate to present these findings at a dedicated venue such as a workshop. We hope the reviews are useful to improve the manuscript.


**Summary Of Ac-Reviewer Meeting:**

A discussion for this paper was attempted. The paper didn’t get any reviewer to champion its acceptance. One of the reviewers shared their post rebuttal conclusions, and recommended a rejection. All arguments are included in the list of concerns 1)-3) above.